EMBO
Molecular Medicine

# Tumor cell-specific inhibition of MYC function using small molecule inhibitors of the HUWE1 ubiquitin ligase

Stefanie Peter[1], Jennyfer Bultinck[2], Kevin Myant[3], Laura A Jaenicke[1], Susanne Walz[1], Judith Müller[4], Michael Gmachl[5], Matthias Treu[5], Guido Boehmelt[5], Carsten P Ade[1], Werner Schmitz[1], Armin Wiegering[6], Christoph Otto[6], Nikita Popov[1,7], Owen Sansom[3], Norbert Kraut[5] & Martin Eilers[1,7,*]

## Abstract

Deregulated expression of MYC is a driver of colorectal carcinogenesis, necessitating novel strategies to inhibit MYC function. The ubiquitin ligase HUWE1 (HECTH9, ARF-BP1, MULE) associates with both MYC and the MYC-associated protein MIZ1. We show here that HUWE1 is required for growth of colorectal cancer cells in culture and in orthotopic xenograft models. Using high-throughput screening, we identify small molecule inhibitors of HUWE1, which inhibit MYC-dependent transactivation in colorectal cancer cells, but not in stem and normal colon epithelial cells. Inhibition of HUWE1 stabilizes MIZ1. MIZ1 globally accumulates on MYC target genes and contributes to repression of MYC-activated target genes upon HUWE1 inhibition. Our data show that transcriptional activation by MYC in colon cancer cells requires the continuous degradation of MIZ1 and identify a novel principle that allows for inhibition of MYC function in tumor cells.

**Keywords** colorectal cancer; HUWE1; MIZ1; MYC; ubiquitination
**Subject Categories** Cancer; Digestive System; Pharmacology & Drug Discovery

See also: **FX Schaub & JL Cleveland** (December 2014)

## Introduction

With over 660,000 new cases each year, colorectal cancer is the most common gastrointestinal malignancy (Jemal *et al*, 2011).

Sequence analysis of tumor genomes shows that each tumor harbors multiple mutations that alter the function of central signaling pathways, which control the growth of colon epithelial cells (CancerGenomeAtlasNetwork, 2012). Predominant among these are the WNT and the RAS signal transduction pathways, and recurrent genetic alterations in both pathways occur in the majority of colorectal cancers (CancerGenomeAtlasNetwork, 2012). Comparison with gene expression profiling shows that enhanced expression and function of the MYC oncoprotein, a downstream effector of both WNT- and RAS-dependent signal transduction, is a common denominator of a vast majority of colon tumors (CancerGenomeAtlasNetwork, 2012; van de Wetering *et al*, 2002). Deletion of MYC ablates tumorigenesis in mouse models of colorectal cancer demonstrating that MYC function is essential for colorectal tumorigenesis (Sansom *et al*, 2007).

MYC is a transcription factor that establishes a gene expression program characteristic of colon epithelial stem cells (Dang, 2012; van de Wetering *et al*, 2002). Like many other transcription factors, MYC is rapidly turned over by the ubiquitin/proteasome system (Gregory & Hann, 2000; Welcker *et al*, 2004). At least three ubiquitin ligases, SKP2, HUWE1 (HECTH9/MULE/ARF-BP1), and FBXO28, are also required for MYC function (Adhikary *et al*, 2005; Cepeda *et al*, 2013; Kim *et al*, 2003; von der Lehr *et al*, 2003). The use of mutants of MYC, in which groups of lysines have been replaced by arginines, suggests that MYC itself needs to be ubiquitinated and possibly degraded to regulate transcription (Adhikary *et al*, 2005; Zhang *et al*, 2013). It is also possible that MYC recruits ubiquitin ligases to degrade repressor proteins that antagonize MYC function. One example for this is the identification of HDAC2, a histone deacetylase that associates with the MXD/MAD complex (Laherty *et al*, 1997) as a substrate for the HUWE1 ligase (Zhang *et al*, 2011).

1  Theodor Boveri Institute, Biocenter, University of Würzburg, Würzburg, Germany
2  Cytokine Receptor Lab, Department of Biochemistry, Ghent University, Ghent, Belgium
3  CRUK Beatson Institute, Glasgow, UK
4  Department of Molecular Oncology, Netherlands Cancer Institute, Amsterdam, the Netherlands
5  Department Lead Discovery, Boehringer Ingelheim RCV GmbH & Co KG, Vienna, Austria
6  Department of General, Visceral, Vascular and Paediatric Surgery, University Hospital Würzburg, Würzburg, Germany
7  Comprehensive Cancer Center Mainfranken, University of Würzburg, Würzburg, Germany
   *Corresponding author. Tel: +49 931 318 4111; E-mail: martin.eilers@biozentrum.uni-wuerzburg.de

In human and mouse tumor cells, MYC binds to target promoters either as part of a binary activating complex with a partner protein, MAX, or as a ternary repressive complex that contains in addition the zinc finger protein MIZ1; the balance of both complexes at each promoter determines the transcriptional response to MYC (Eilers & Eisenman, 2008; Walz *et al*, 2014). HUWE1 associates with MYC, the related N-MYC protein, and with MIZ1 and ubiquitinates all three proteins (Adhikary *et al*, 2005; Li *et al*, 2008; Yang *et al*, 2010). Ubiquitination by HUWE1 degrades both N-MYC and MIZ1 and restricts N-MYC function *in vivo* (Zhao *et al*, 2009). In contrast, HUWE1 has only weak effects on the turnover of MYC (Adhikary *et al*, 2005; Zhao *et al*, 2008). One possibility is that HUWE1 assembles K63-linked ubiquitin chains on MYC that do not target the protein for degradation (Adhikary *et al*, 2005). Alternatively, lysines that are critical for degradation of MYC may be buried (e.g., by complex formation with MIZ1) and therefore inaccessible to HUWE1 *in vivo* (Kim *et al*, 2011).

The effects of HUWE1 depletion or deletion on the function of MYC proteins have been characterized in several biological contexts: Deletion of HUWE1 enhances N-MYC levels in embryonic stem cells cultured in the absence of leukemia inhibitory factor and expands neuronal cell population in the cerebellum (D'Arca *et al*, 2010; Zhao *et al*, 2009). In skin papillomas, deletion of HUWE1 stabilizes MIZ1 and enhances repression of *Cdkn2b* (p15INK4b) and *Cdkn1a* (p21CIP1) by the MYC/MIZ1 complex, correlating with enhanced tumorigenesis (Inoue *et al*, 2013). In contrast, depletion of HUWE1 in several human tumor cells arrests proliferation and inhibits expression of MYC-activated target genes (Adhikary *et al*, 2005). Here, we used both shRNAs and small molecule inhibitors to explore the role of HUWE1 as a regulator of MYC function in human colon cancer cells. The aim of this study was to test the hypothesis that inhibition of HUWE1 is feasible to control MYC activity in this tumor entity and to explore the underlying mechanistic basis.

## Results

To test whether HUWE1 is required for growth of colon cancer cells in culture, we generated retroviruses that constitutively express shRNAs targeting HUWE1. We identified two shRNA sequences that strongly reduced expression of HUWE1 as determined by RQ-PCR and immunoblotting of lysates of infected cells (Fig 1A, left and middle panel). Expression of either shRNA suppressed the clonogenic growth of Ls174T colon carcinoma cells, which depend on MYC for proliferation (van de Wetering *et al*, 2002), relative to control-infected cells (Fig 1A, right panel). Depletion of HUWE1 also suppressed the growth of three other colon carcinoma cell lines that we tested (Supplementary Fig S1A and B). To rule out that the effects on colony formation were due to variations in infection efficiency, we generated Ls174T cells that express one of the two shRNAs targeting HUWE1 in a doxycycline-inducible manner (Fig 1B). Consistent with the results obtained upon constitutive expression of shHUWE1, addition of doxycycline led to a depletion of HUWE1 and strongly suppressed colony formation (Fig 1C). These results were confirmed with a second doxycycline-regulated shRNA targeting HUWE1 (shHUWE1-3) (Fig 1B and C).

We injected Ls174T cells expressing both inducible shRNAs targeting HUWE1 subcutaneously in immunocompromised mice.

Addition of doxycycline to the drinking water suppressed expression of HUWE1 and strongly decreased BrdU incorporation in subcutaneous tumors (Fig 1D). Addition of doxycycline retarded (shHUWE1-2; Supplementary Fig S2A) or abrogated (shHUWE1-3) subcutaneous tumor growth (Fig 1E and Supplementary Fig S2B). We also tested the effect of depletion of HUWE1 in an orthotopic setting, in which Ls174T cells stably expressing luciferase and doxycycline-inducible shHUWE1-3 were transplanted into the cecum of immunocompromised mice. Tumor growth was monitored twice a week by luciferase-based *in vivo* imaging. Out of 12 grafted mice, six developed a primary tumor in the colon. Half of these mice were left untreated, resulting in outgrowth of the primary tumor and their subsequent dissemination to the peritoneum, lymph nodes, liver, and lung. Addition of doxycycline strongly suppressed the growth of tumors in this orthotopic setting (note the logarithmic scale) and suppressed the formation of metastases (Fig 1F; data for individual mice are shown in Supplementary Fig S2C). We concluded that HUWE1 is required for growth and tumor formation of human colon cancer cells.

To understand the mechanisms underlying these observations, we isolated RNA from pools of Ls174T cells stably expressing shRNA targeting HUWE1. Immunoblots showed that depletion of HUWE1 had no significant effect on steady-state levels of MYC (Fig 2A), consistent with previous observations (Adhikary *et al*, 2005). RQ-PCR assays revealed that constitutive depletion of HUWE1 downregulated multiple genes that are activated by MYC relative to control-infected cells (Fig 2B). In contrast, knockdown of HUWE1 had only marginal effects on expression of a control gene (*ACTB*) or on expression of two genes, *CDKN1A* and *GADD45A*, that are repressed by MYC (Fig 2B). Depletion of HUWE1 also strongly increased expression of *MUC2*, a gene that is induced during terminal differentiation of colon epithelial cells (Fig 2B); this is consistent with the observation that MYC suppresses differentiation of Ls174T cells (van de Wetering *et al*, 2002). Virtually identical changes in gene expression were observed in Ls174T cells expressing a doxycycline-inducible shRNA targeting HUWE1 (Fig 2C). To explore the changes in gene expression elicited by depletion of HUWE1 in an unbiased manner, we performed microarray analyses and found that depletion of HUWE1 led to upregulation of 492 and downregulation of 250 genes, respectively (cut-off: fold change 2; $P < 0.01$). Gene set enrichment analysis (GSEA) showed that multiple sets of MYC-activated genes were downregulated upon depletion of HUWE1, and this was statistically highly significant (Fig 2D, upper panel) (Subramanian *et al*, 2005). In contrast, sets of MYC-repressed genes were not significantly affected, arguing that HUWE1 is required for activation, but not repression by MYC (Fig 2D, lower panel). The sets of MYC-activated genes include many target genes of MYC, like *ODC1* or *HSPE1*, that are not regulated in a cell cycle-dependent manner, demonstrating that downregulation of these genes is not an indirect consequence of the cell cycle arrest induced by depletion of HUWE1. HUWE1 also ubiquitinates p53 and promotes the degradation of p53 (Chen *et al*, 2005), but depletion of HUWE1 had no significant effects on steady-state levels of p53 or phosphorylation of ATM or CHK2, in contrast to cells exposed to DNA damage (Fig 2E). Furthermore, shRNA-mediated depletion of p53 did not alleviate the effects of shHUWE1 on expression of MYC target genes (Supplementary Fig S3A and B). We concluded that HUWE1 is required for activation, but not repression, of MYC target genes in colon carcinoma cells.

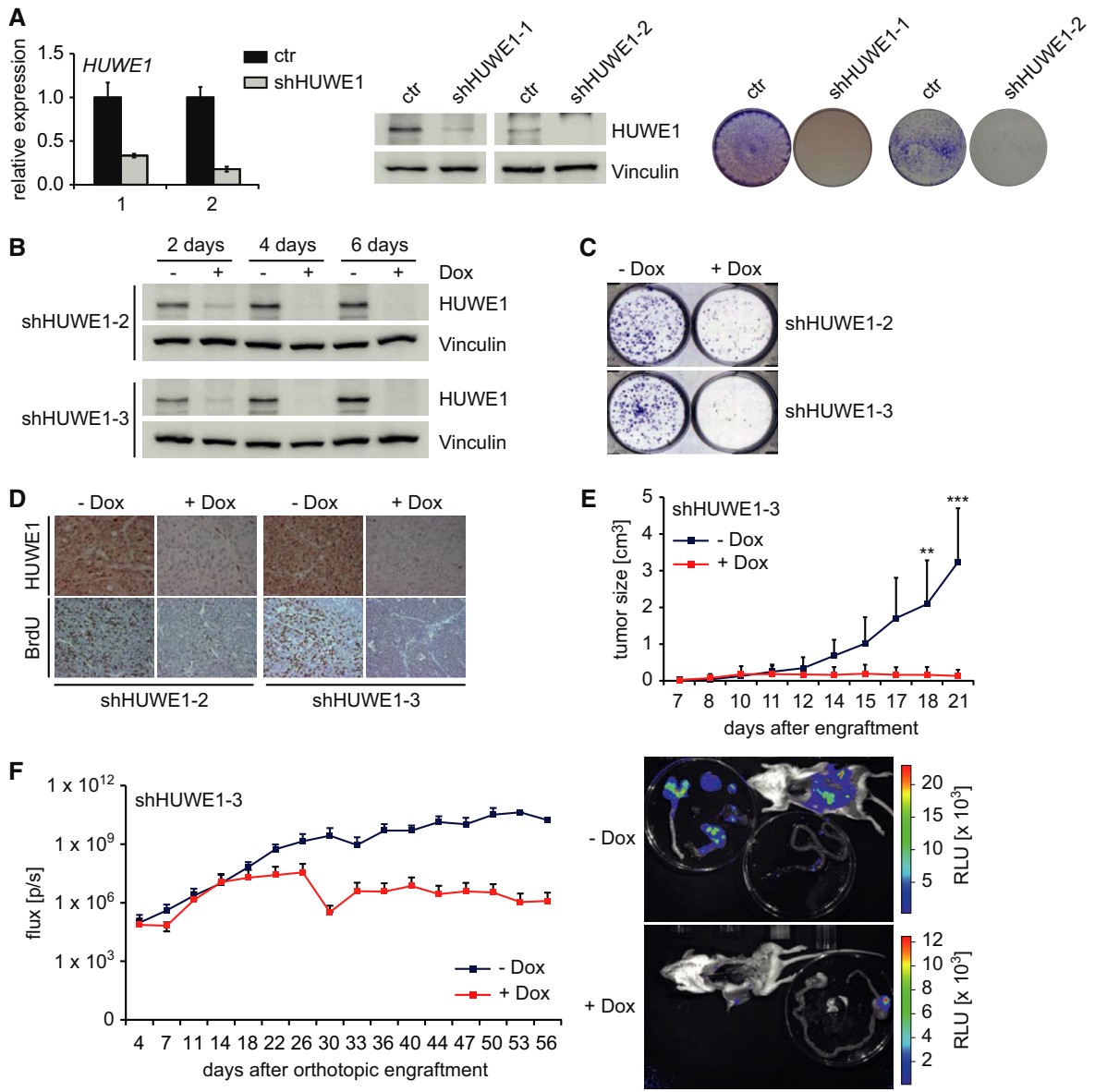

**Figure 1. Effect of HUWE1 depletion on growth and tumor formation of colorectal cancer cells.**

A  The left panel shows RQ–PCR assays documenting expression of *HUWE1* in Ls174T cells after infection with either control or two different shRNAs targeting HUWE1. The middle panel shows an immunoblot documenting expression of HUWE1 in these cells. Vinculin was used as loading control. The right panel shows a colony formation assay of Ls174T cells expressing control or shRNAs targeting HUWE1 (*n* = 3; unless otherwise indicated, *n* indicates the number of animals or independent biological repeat experiments in the following legends).

B  Ls174T cells expressing doxycycline-inducible shRNAs targeting HUWE1 were treated with 1 µg/ml doxycycline or solvent control. Cells were harvested at the indicated time points, and immunoblots of the lysates were probed with antibodies against HUWE1 or Vinculin as loading control.

C  Colony formation assay of Ls174T cells expressing doxycycline-inducible shRNAs targeting HUWE1. Cells were cultured for 10 days in the presence or absence of 1 µg/ml doxycycline as indicated. Colonies were stained with crystal violet.

D  Immunohistochemistry documenting expression of HUWE1 and incorporation of BrdU in subcutaneous tumors of Ls174T cells expressing doxycycline-inducible shRNAs targeting HUWE1. Seven days after start of doxycycline treatment, mice were injected with BrdU. 1 h after injection, tumors were dissected for immunostaining.

E  Subcutaneous tumor growth during doxycycline treatment. The experiment was performed as described in (D). Seven days after engraftment, tumors became palpable and doxycycline treatment was started. Tumor growth was followed for 2 weeks. Data are presented as mean + standard deviation (SD) (*n* = 5). *P*-values were calculated using Student's *t*-test (***P*-value < 0.01, ****P*-value < 0.001).

F  Growth of orthotopic Ls174T tumors grafted in the cecum of immunocompromised mice during doxycycline treatment. Doxycycline treatment was started when all the mice showed an abdominal luciferase signal (14 days after engraftment), and tumor growth was followed for six weeks (left panel). Data are represented as mean + SD (*n* = 3 for most time points, see Supplementary Fig S2C). At the end of the experiment, mice were dissected and luciferase activity was assessed in different organs. One example is shown in the right panel. flux [p/s] = photons per sec. RLU = relative light units.

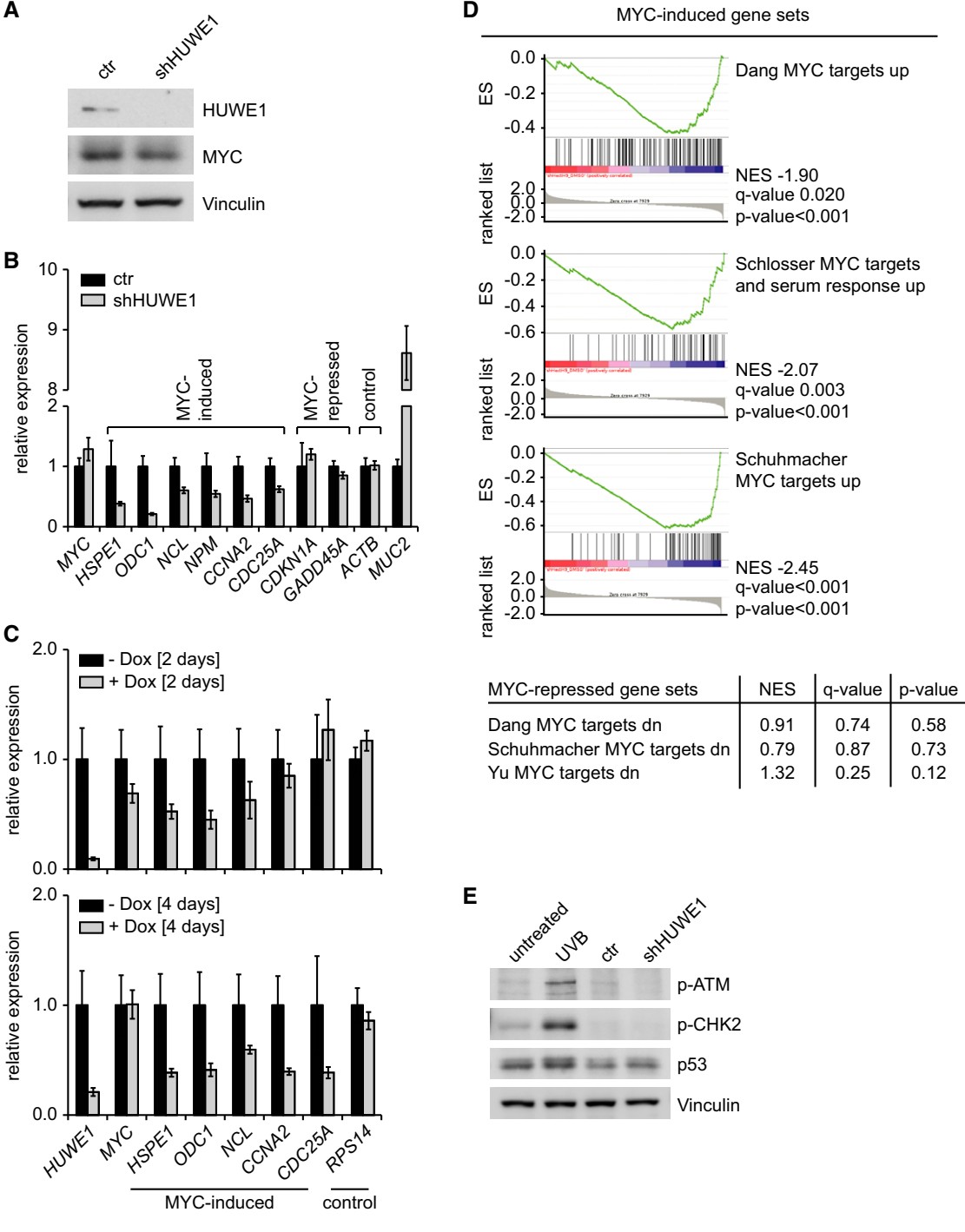

**Figure 2.  Effect of HUWE1 depletion on expression of MYC target genes.**

A   Expression of MYC in HUWE1-depleted Ls174T cells. Ls174T cells were stably infected with retroviruses expressing shRNA targeting HUWE1. After selection, cells were harvested and immunoblots of cell lysates were probed with indicated antibodies.

B   RQ–PCR assays documenting expression of the indicated genes in Ls174T cells expressing control shRNA or shRNA targeting HUWE1. Error bars show SD of triplicate technical assays from one representative experiment (*n* = 5).

C   RQ–PCR assays documenting expression of the indicated genes in Ls174T cells expressing doxycycline-inducible shRNA targeting HUWE1 at two different time points after addition of doxycycline. Error bars show SD of triplicate technical assays from one representative experiment (*n* = 2).

D   Effects of HUWE1 knockdown on expression of previously identified sets of MYC target genes. The upper panel displays results of a gene set enrichment analysis (GSEA), in which known sets of MYC-induced genes from the Molecular Signature Database (MSigDB) are compared to genes regulated by HUWE1 depletion in Ls174T cells. The table below shows examples of MYC-repressed gene sets.

E   Effects of HUWE1 knockdown on p53 levels and DNA damage. The experiment was performed as described in (A). As control, Ls174T cells were left untreated or exposed to 500 J/m² UVB and harvested 3 h later.

To identify potential inhibitors of HUWE1, we configured an *in vitro* assay of HUWE1 activity for high-throughput screening of small molecules, exploiting the fact that the HECT-domain of HUWE1 auto-ubiquitinates (Pandya *et al*, 2010). Briefly, the HECT-domain of HUWE1 was biotin-tagged, attached to streptavidin-coated 96-well plates, and incubated with UBA1, UbcH5b, a ubiquitin-conjugating enzyme that supports HUWE1 activity *in vitro* (Adhikary *et al*, 2005), ATP, and MYC-tagged ubiquitin. Auto-ubiquitination was detected using a europium-labeled MYC antibody and appropriate detection reagents (Fig 3A). Screening of 840,243 compounds resulted in 2,765 hits that inhibited HUWE1 activity, yielding an initial hit rate of 0.33%. After hit confirmation in repeat experiments, we identified inhibitors of UBA1 or UbcH5b by a thioester assay that measures the covalent binding of UbcH5b to ubiquitin in the presence of UBA1: Compounds active in this assay were eliminated. Furthermore, dose responses were determined in both HUWE1 and NEDD4 auto-ubiquitination assays, and compounds inhibiting NEDD4 auto-ubiquitination were eliminated (Fig 3B and unpublished observations). From these experiments, we selected two compounds (BI8622 and BI8626) that inhibited HUWE1 with $IC_{50}$ values of 3.1 μM (BI8622) and 0.9 μM (BI8626), respectively (Fig 3B and C). HECT-domains of multiple ligases auto-ubiquitinate in *in vitro* assays containing both UBA1 and UbcH5b (M. Gmachl, unpublished observation). These assays were used to analyze the specificity of the identified inhibitors. We found that neither compound inhibited the activity of other HECT-domain ubiquitin ligases in these assays, arguing that they are specific inhibitors of HUWE1 (Fig 3C). Attempts to co-crystallize compound/HUWE1 complexes failed due to the very high solubility of the HECT-domain of HUWE1 (M. Gmachl, unpublished observation).

To test the efficacy of both compounds in tissue culture, we initially confirmed observations that HUWE1 ubiquitinates and degrades MCL1 in response to DNA damage (Zhong *et al*, 2005). Consistent with these published data, steady-state levels of MCL1 decreased rapidly upon UV irradiation of U2OS cells and depletion of HUWE1 both enhanced steady-state levels of MCL1 and retarded the decrease upon UV irradiation (Supplementary Fig S4A). Furthermore, ectopically expressed HUWE1 ubiquitinated MCL1 (Supplementary Fig S4B) and ubiquitination of MCL1 increased after UV irradiation (Supplementary Fig S4C) in a HUWE1-dependent manner (Supplementary Fig S4D). These data confirmed that degradation of MCL1 upon UV irradiation is a valid read-out of HUWE1 activity. Incubation of HeLa cells with either HUWE1 inhibitor abolished ubiquitination of MCL1 induced by ectopically expressed HUWE1 (Fig 3D). For BI8622, we performed multiple ubiquitination assays in the presence of different concentrations of inhibitor; this yielded an $IC_{50}$ value of 6.8 μM, similar to that obtained *in vitro* (Supplementary Fig S4E). Both compounds retarded the degradation of MCL1 in response to UV irradiation to the same extent as depletion of HUWE1 (Fig 3E). Furthermore, both compounds induced accumulation of TopBP1 (Fig 3F), another substrate of HUWE1 (Herold *et al*, 2008). Consistent with the observed effects in response to depletion of HUWE1, neither compound led to an accumulation of p53 in Ls174T cells (Supplementary Fig S5A). We concluded that both BI8622 and BI8626 inhibit HUWE1 in cells.

Incubation with HUWE1 inhibitors suppressed colony formation of Ls174T cells with estimated $IC_{50}$ values of 8.4 μM (BI8622) and 0.7 μM (BI8626), respectively, which is in good agreement with the $IC_{50}$ values for inhibition of HUWE1 activity (Fig 4A). Both inhibitors also suppressed the growth of three additional colon carcinoma cell lines that we tested with minor variations in sensitivity (Supplementary Fig S5B). Partial shRNA-mediated depletion of HUWE1 led to a moderate, but highly significant reduction in $IC_{50}$ value for growth inhibition by BI8622, arguing that growth inhibition is an on-target effect of the inhibitor (Supplementary Fig S6). FACS assays of propidium iodide-stained cells showed that both compounds led to a decrease in the percentage of cells in the S and G2 phases and an accumulation of Ls174T cells in the G1 phase of the cell cycle, but had only marginal effects on apoptosis (Fig 4B). Combining these data with a growth curve (Fig 4C) enabled us to calculate the length of each phase of the cell cycle (Supplementary Fig S7A). Incubation with either compound retarded passage of Ls174T cells through all phases of the cell cycle, with the effect being strongest for G1. In contrast to colon carcinoma cells, HUWE1 is not required for proliferation of embryonic stem (ES) cells and deletion of HUWE1 instead delays the decrease of N-MYC levels that occurs upon removal of leukemia inhibitory factor (LIF) from the medium (Zhao *et al*, 2008). Consistent with these data, chemical inhibition of HUWE1 had marginal effects on the growth and cell cycle distribution of ES cells as well as expression of MYC target genes in the presence of LIF and retarded the decrease in N-MYC levels when LIF was withdrawn (Fig 4D–F and Supplementary Fig S7B). Furthermore, inhibition of HUWE1 in crypt cultures of normal intestinal epithelial cells had no effect on expression of MYC target genes (Fig 4G). We concluded that both compounds inhibit MYC-dependent transactivation in colon cancer cells, but not in normal

---

**Figure 3.  Identification of small molecule inhibitors of HUWE1.**

A   Scheme of the high-throughput screen for identification of small molecule HUWE1 inhibitors.

B   The upper panels show the structures of BI8622 and BI8626. The lower panel displays the corresponding $IC_{50}$ curves of compounds BI8622 and BI8626. The curves are presented as mean ± SD ($n = 2$).

C   Table of the $IC_{50}$ values for BI8622 and BI8626 in *in vitro* ubiquitination assays with HUWE1 and other HECT E3 ubiquitin ligases.

D   *In vivo* ubiquitination of MCL1. HeLa cells were transfected with plasmids expressing MCL1, His-ubiquitin, and HA-HUWE1. Where indicated, cells were treated with compounds BI8622 and BI8626 or DMSO as solvent control. After lysis, ubiquitin-modified proteins were recovered by Ni-NTA-Agarose and immunoblots of the eluates were probed with an antibody against MCL1. The input control shows 1% of the total lysate.

E   UV-induced degradation of MCL1 after HUWE1 inhibition. U2OS cells were treated with BI8622 (upper panel) or BI8626 (lower panel) or DMSO as control. At the same time, cells were irradiated with 500 J/m² UVB and harvested after indicated time points. Immunoblots of the lysates were probed with antibodies recognizing either MCL1 or CDK2 as control. Quantification of the MCL1 protein levels for the according experiment is shown directly below the immunoblots. Diagrams below the immunoblots show mean protein levels ± SD ($n = 5$). From the data, the half-life of the MCL1 protein was determined with *P*-values calculated using a Student's *t*-test.

F   Effects of HUWE1 inhibition on levels of TopBP1. Ls174T cells were treated with BI8622 and BI8626 or DMSO (−) as control for the indicated time points. Cell lysates were fractionated in chromatin-bound and soluble non-chromatin-bound fractions. Immunoblots of the soluble fraction were incubated with indicated antibodies.

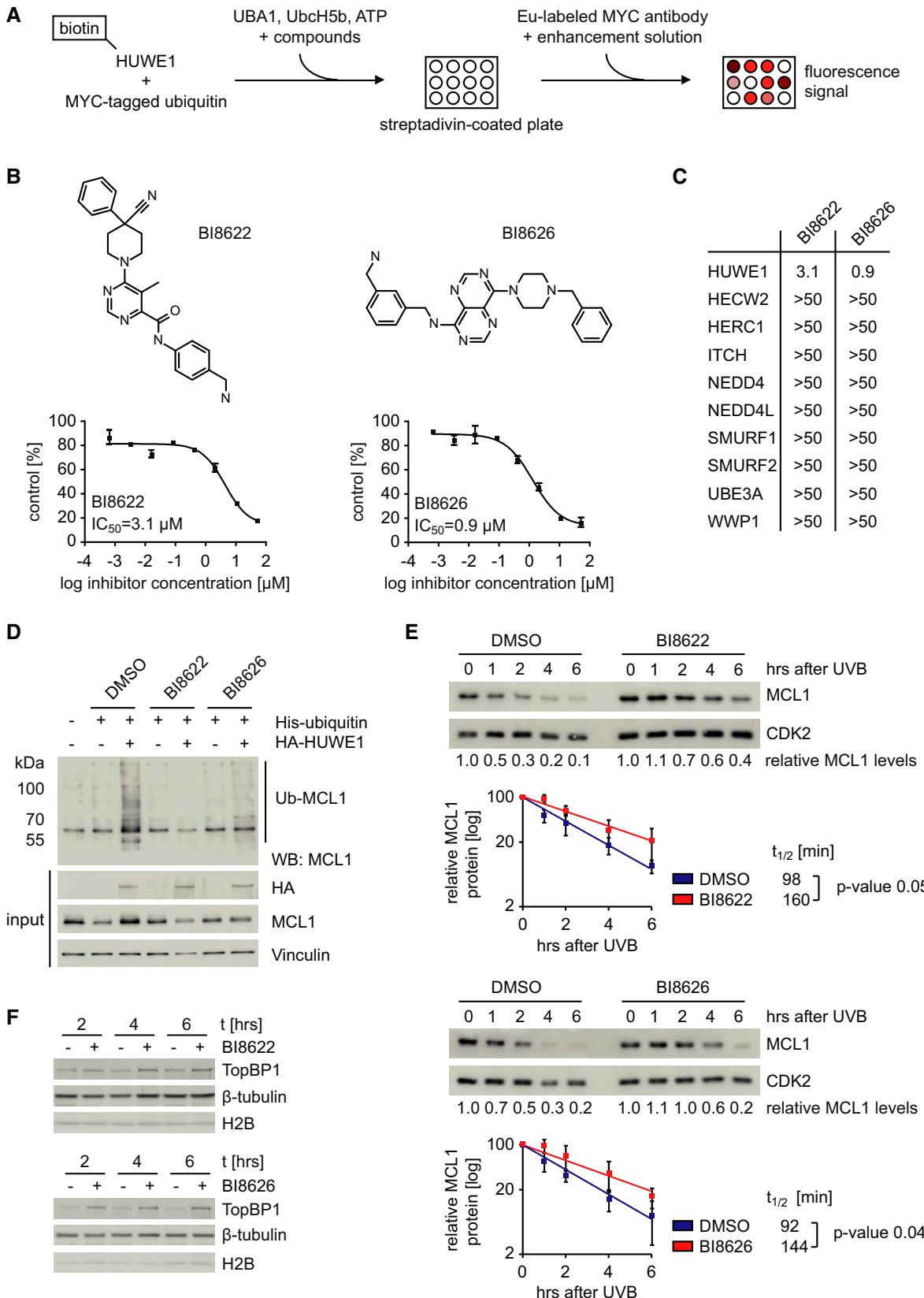

Figure 3.

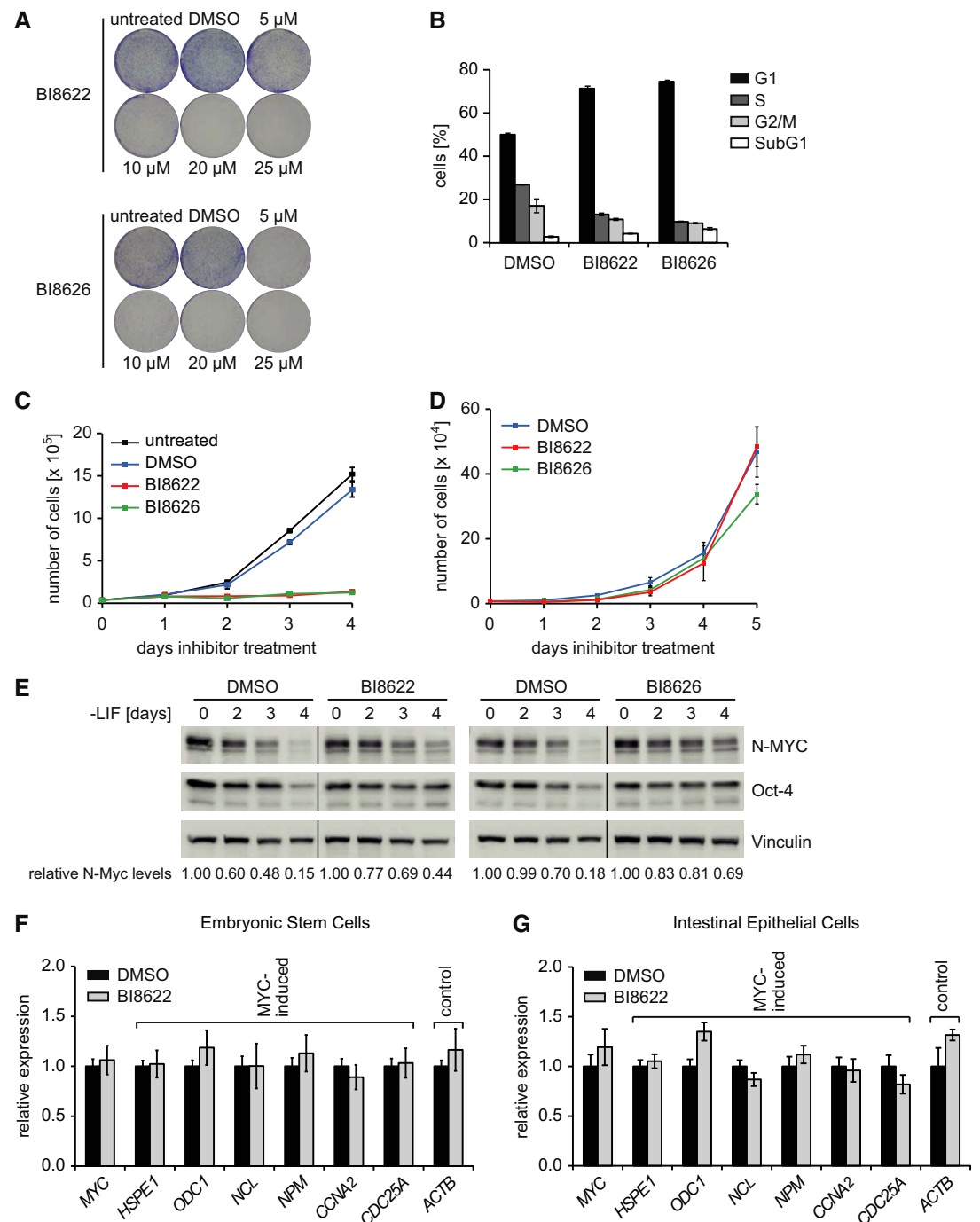

**Figure 4. Effect of HUWE1 inhibition on growth and gene expression in epithelial and embryonic stem cells.**

A   Colony formation assays documenting growth of Ls174T cells in the presence of the indicated concentrations of HUWE1 inhibitors. Cells were stained after 5 days.

B   Cell cycle distribution of Ls174T cells upon HUWE1 inhibition. Inhibitors or DMSO as control were added to Ls174T cells for 24 h. Before harvesting, cells were labeled for 1 h with BrdU. Error bars represent SD of technical triplicates from one representative experiment ($n = 2$).

C   Growth curve of Ls174T cells cultivated in the presence of HUWE1 inhibitors. Untreated or DMSO-treated samples served as control.

D   Proliferation of HUWE1 inhibitor-treated murine embryonic stem (ES) cells. ES cells were cultivated in the presence of leukemia inhibitory factor (LIF) and HUWE1 inhibitors (10 μM) or DMSO as control. Data show mean cell number ± SD ($n = 3$).

E   ES cells were deprived of LIF for the indicated times and in parallel treated with HUWE1 inhibitors or DMSO as control. The panel shows immunoblots documenting N-MYC, Oct-4, and Vinculin protein levels. A quantitation of N-MYC protein levels relative to Vinculin is shown below the blot ($n = 3$).

F, G   RQ–PCR assays documenting expression of MYC target gene expression in murine ES cells (F) and in crypt cultures of normal intestinal epithelial cells (G) upon inhibition of HUWE1. ES cells were cultivated in the presence of LIF and treated with BI8622 (10 μM) for 48 h (left panel). Intestinal organoid cultures were treated with BI8622 (10 μM) for 24 h (right panel). DMSO served as solvent control. Error bars show SD of technical triplicate assays of one representative experiment (F: $n = 2$; G: $n = 3$).

intestinal epithelial cells and embryonic stem cells. *In vitro* assays revealed that both compounds are unstable in the presence of microsomes (Supplementary Fig S7C). Measurements of compound levels in serum after intraperitoneal injection in mice showed that neither compound accumulated to high levels and both were rapidly cleared after injection, precluding a more detailed *in vivo* analysis of the efficacy of these compounds (Supplementary Fig S7D).

To test whether the compounds inhibit transactivation of MYC, we infected Ls174T cells with retroviruses expressing either control shRNA or shRNA targeting HUWE1 and incubated pools of stably infected cells with either compound or DMSO as control for 24 h. Both inhibitors reduced the expression of several MYC target genes in control cells, but had no effect in HUWE1-depleted cells (Fig 5A). Furthermore, inhibition of HUWE1 resulted in a strong increase in expression of *MUC2* (Fig 5B). Microarray analyses showed that both compounds led to down- and upregulation of multiple genes (BI8622: 2,267 up, 2,295 down; BI8626: 2,796 up, 2,923 down; cutoff: fold change 2; $P < 0.01$). Gene set enrichment analysis showed that both compounds downregulated virtually identical gene sets (Fig 5C). In contrast, upregulated genes were contained in a much smaller number of previously known gene sets and these differed between compounds (Fig 5C). This suggested that the downregulation of genes most likely corresponds to the on-target activity of either chemical. Direct comparisons showed that the effects of both compounds on gene expression were highly similar to depletion of HUWE1 (Fig 5D). Importantly, GSEA showed that expression of multiple sets of MYC-activated genes was significantly repressed upon exposure to either compound, whereas MYC-repressed genes were not consistently affected (Fig 5E and F). As described above, analysis of candidate genes showed that both inhibitors have little effect on expression of MYC target genes in HUWE1-depleted cells, arguing that their effect is mediated via inhibition of HUWE1 (Fig 5A). A genome-wide analysis of gene expression confirmed that BI8622 had no statistically significant effect on expression of any set of MYC-activated target genes in HUWE1-depleted cells, in contrast to its effects in control cells (Fig 5G). Finally, shRNA-mediated depletion of p53 (Supplementary Fig S8, upper panel) had no effect on regulation of MYC target genes by either compound (Supplementary Fig S8, lower panel). Collectively, the data establish that these compounds block MYC-dependent transcriptional activation via inhibition of HUWE1 in colon cancer cells.

Neither depletion nor chemical inhibition of HUWE1 affected the expression of any member of the MXD family of repressor proteins that antagonize transactivation by MYC or nuclear localization of MYC (Supplementary Figs S9A and S10). Furthermore, inhibition of HUWE1 did not inhibit complex formation of MYC with MAX (Supplementary Fig S9B), raising the question how HUWE1 affects MYC-dependent transactivation. Consistent with published data, ectopic expression of HUWE1 strongly decreased steady-state levels of MIZ1 (Fig 6A), a zinc finger protein with which MYC forms a repressive complex and a previously identified target of HUWE1 (Inoue *et al*, 2013; Walz *et al*, 2014; Wiese *et al*, 2013; Yang *et al*, 2010). In contrast, ectopic expression of HUWE1 had only moderate effects on steady-state levels of MYC, consistent with the results obtained in response to depletion of HUWE1 (Fig 6A). As a result, ectopic expression of HUWE1 increased the relative amount of MYC that is not bound to MIZ1 (Fig 6B). Notably, HUWE1 also decreased levels of MIZ1 in the presence of a MYC protein devoid of all lysine residues, suggesting that ubiquitination of MYC itself may not strictly be required for its effects on MYC-dependent gene regulation (Fig 6C). Inhibition of HUWE1 blocked the HUWE1-mediated decrease in MIZ1 levels; the decrease was also blocked by incubation of cells with an inhibitor of proteasomal degradation, MG132 (Supplementary Fig S11A). Consistently, depletion or chemical inhibition of HUWE1 increased steady-state levels of MIZ1 in colon carcinoma cells (Fig 6D and E). Similarly, depletion or inhibition of HUWE1 increased levels of MIZ1 in mouse keratinocytes (Supplementary Fig S11B and C), in which HUWE1 is known to degrade MIZ1 (Inoue *et al*, 2013).

Consistent with the immunoblot data, ChIP-sequencing experiments showed that inhibition of HUWE1 strongly increased the amount of MIZ1 bound to MYC-bound core promoters (Fig 6F–H and Supplementary Fig S12). Inhibition of HUWE1 also led to a very small increase in MYC binding, consistent with observations that complex formation with MIZ1 stabilizes MYC (Salghetti *et al*, 1999). Depletion or inhibition of HUWE1 induced a significant decrease in acetylation of histone H3 at the *HSPE1* promoter, but not at a control (*ACTB*) promoter (Supplementary Fig S13A and B). In contrast, inhibition of HUWE1 had only weak effects on expression of direct target genes of MIZ1 (Supplementary Fig S13C) and on histone acetylation at a directly MIZ1-bound promoter (*VAMP4*) (Supplementary Fig S13B). The data argue that inhibition of

**Figure 5. Effect of HUWE1 inhibition on expression of MYC target genes.**

A   Expression of MYC target genes after HUWE1 inhibition in control and HUWE1-depleted Ls174T cells. Control or HUWE1-depleted Ls174T cells were treated with HUWE1 inhibitors or DMSO as control for 24 h. Expression of different MYC-induced target genes was analyzed by RQ–PCR. Error bars show SD of triplicate technical assays from one representative experiment ($n = 5$).

B   Time-dependent expression of the differentiation marker *MUC2* upon HUWE1 inhibition. Ls174T cells were treated with HUWE1 inhibitors for the indicated times. Error bars show SD of triplicate technical assays from one representative experiment ($n = 3$).

C   Gene set enrichment analysis (GSEA) of gene expression changes upon HUWE1 inhibition. Ls174T cells were treated for 24 h with either HUWE1 inhibitor or DMSO as control. RNA was isolated and subjected to a microarray and gene set enrichment analysis. The Venn diagrams display the number of gene sets that overlap between BI8622 and BI8626 treated cells for down- and upregulated genes. *P*-values were calculated using a test for hypergeometric distribution.

D   Heat map comparing gene expression after HUWE1 depletion to treatment with HUWE1 inhibitors. The left panel summarizes expression of all genes ($n = 10,452$) ordered according to their expression after HUWE1 knockdown. The right panel shows all genes that have a MYC binding site in the promoter (defined as −1 kb to +0.5 kb relative to the TSS) ($n = 5,664$). MYC binding data were taken from HeLa cells. *P*-values for correlation were calculated using a Spearman's rank test.

E   Examples of sets of genes that are downregulated upon HUWE1 inhibition. The panel shows examples of sets of MYC-activated genes that are repressed upon incubation with either HUWE1 inhibitor. Additional signatures are examples of genes related to MYC function.

F   Effects of HUWE1 inhibition on expression of MYC target gene sets. The upper panels display examples of MYC-activated gene sets and their response to inhibition of HUWE1. The table below shows examples of MYC-repressed gene sets and their response to BI8622 and BI8626.

G   The panel shows the response of a representative set of MYC-activated target genes to BI8622 in control and HUWE1-depleted cells.

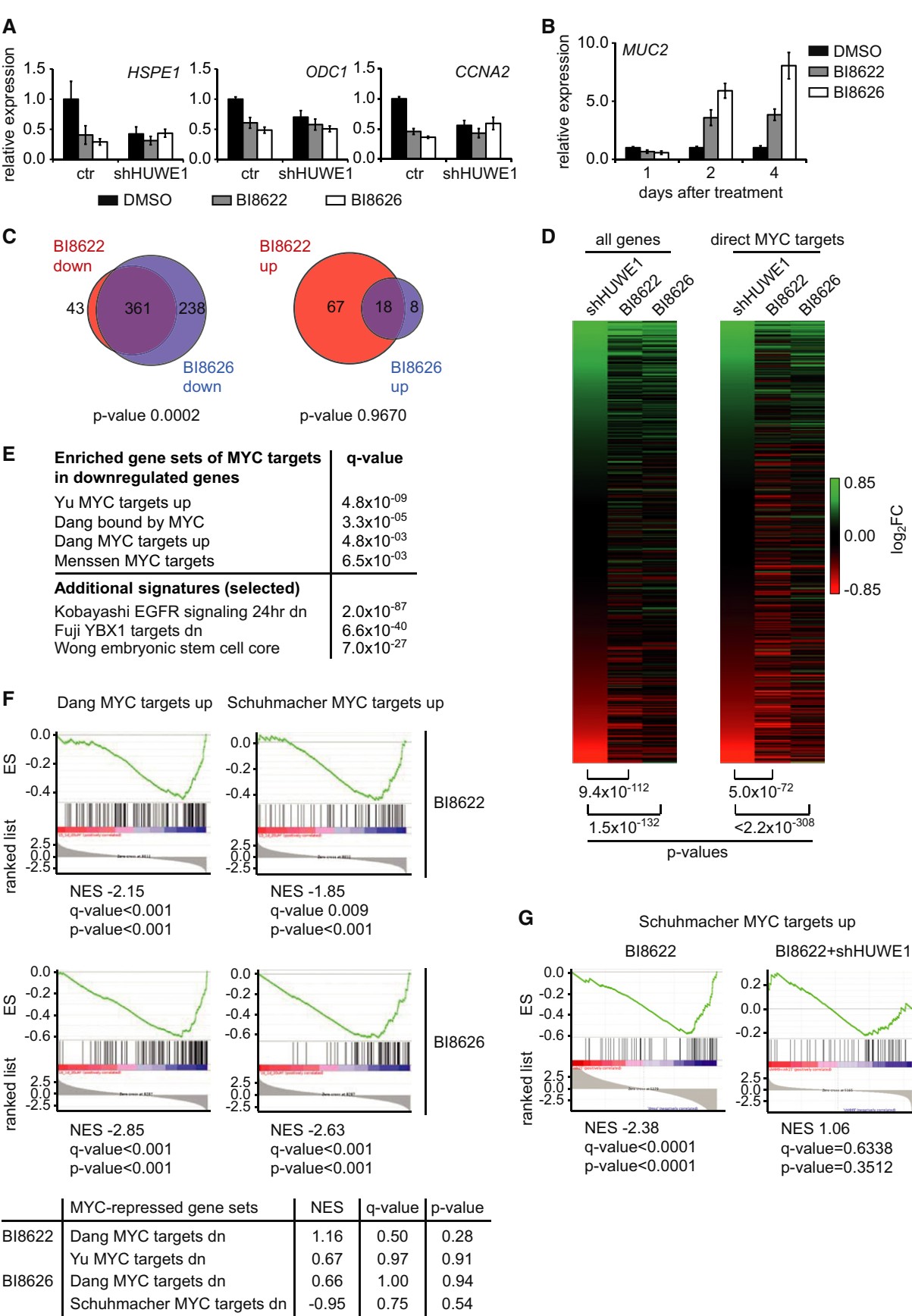

**Figure 5.**

**Figure 6.  Effect of HUWE1 on MYC and MIZ1 complexes.**

A  Immunoblot documenting levels of MYC and MIZ1 upon HUWE1 overexpression. HeLa cells were transfected with plasmids expressing MYC, MIZ1, and HUWE1 (wt or catalytically inactive CS mutant). Relative protein levels of MIZ1 and MYC are shown below the immunoblot (*n* = 3).

B  Interaction of MYC and MIZ1 upon ectopic expression of HUWE1. HeLa cells were transfected with the indicated expression vectors. Cell lysates were immunoprecipitated with MYC antibody or IgG as control, and immunoblots of the eluates were probed with indicated antibodies. As input control, 5% of the lysate was loaded. For each immunoprecipitate, the relative amount of MYC that is not bound to MIZ1 is shown below the blot (*n* = 2).

C  Steady-state protein levels of lysine-free (MYC-K) MYC and MIZ1 upon HUWE1 overexpression. The experiment was performed as described in (A).

D  Immunoblots documenting steady-state protein levels of MYC and MIZ1 upon HUWE1 depletion in Ls174T cells.

E  Steady-state protein levels of MYC and MIZ1 upon HUWE1 inhibition. Ls174T cells were treated with the indicated concentrations of HUWE1 inhibitors or DMSO as control. After 24 h, cells were lysed and immunoblots of the lysates probed with indicated antibodies.

F  Heat map of all human transcription start sites (TSS) (±5 kb) documenting increased MIZ1 binding upon treatment with BI8622 (20 μM) in Ls174T cells. Genes were ranked according to MYC occupancy in DMSO control situation.

G  Tag density distribution of MYC and MIZ1 around human TSS before and after treatment with BI8622. Cells were treated as described in (A). Tags were counted in 50 bp windows, and tags of the respective IgG control were subtracted.

H  Boxplot illustrating recruitment (as $\log_2$FC) of MYC and MIZ1 after BI8622 treatment at MIZ1-bound promoters (−1 kb to +0.5 kb) containing a consensus E-box (CACGTG; *n* = 1711). Cells were treated as described in (A). For analysis, tags were counted in a region ±100 bp around the summit of the MIZ1 peak and *P*-values were calculated using a two-tailed, one-sample *t*-test with μ = 0.

HUWE1 shifts the balance on MYC target genes toward repressive MYC/MAX/MIZ1 complexes thereby inhibiting transactivation by MYC. To test whether accumulation of MIZ1 is required for repression of MYC target genes, we inhibited HUWE1 in control cells and in cells that express a doxycycline-inducible shRNA targeting MIZ1 (Fig 7A). RQ–PCR assays showed that depletion of MIZ1 alleviated repression of individual MYC-activated target genes (Fig 7B). GSEA of RNA sequencing data showed that depletion of MIZ1 did not uniformly relieve repression of MYC-activated target genes by HUWE1 inhibitors; instead, the effect was strongest on target genes encoding ribosomal proteins, in essence abolishing repression by HUWE1 inhibitors (Fig 7C and D). Analysis of ChIP-sequencing data confirmed that MIZ1 accumulated at the core promoter of ribosomal genes upon HUWE1 inhibition (Fig 7E). The data show that HUWE1 is required for transactivation by MYC at least in part since it counteracts the assembly of repressive MYC/MIZ1 complexes at E-box target genes that are bound by MYC (see Fig 7F for a model). We note that histone deacetylase 2, another target that is degraded by HUWE1, has been implicated in transcriptional repression at E-box sites that are transactivated by MYC (see Discussion) (Laherty *et al*, 1997). While effects of HUWE1 on total cellular pools of HDAC2 were small under our experimental conditions (Fig 6A,C,D, E), it is possible that local degradation of HDAC2 by HUWE1 at MYC-bound E-boxes contributes to transcriptional activation by MYC.

## Discussion

A large body of proof-of-principle experiments suggests that targeting MYC proteins may have considerable therapeutic benefit in human tumors (Felsher & Bishop, 1999; Soucek *et al*, 2008). As a consequence, several strategies have been proposed to target MYC or N-MYC using small molecule inhibitors. One strategy is based on a class of small molecules that bind to the leucine zipper of MYC and N-MYC and inhibit heterodimerization with MAX, which is critical for transformation by MYC proteins (Yin *et al*, 2003). A second way of targeting MYC is the use of small molecule inhibitors that interfere with MYC expression or stability: Examples are inhibitors of the bromodomain protein BRD4, which is critical for transcription of the *MYC* promoter, and inhibitors of the Aurora-A kinase that disrupt a

stabilizing interaction of Aurora-A with N-MYC (Brockmann *et al*, 2013; Delmore *et al*, 2011). We propose here that targeting the HUWE1 ubiquitin ligase is feasible and allows for a tumor cell-specific inhibition of MYC function.

We have shown previously that both activating and repressive complexes of MYC can be detected at most MYC-bound promoters in tumor cells (Walz *et al*, 2014; Wiese *et al*, 2013). Previous work had also shown that HUWE1 is a degrading ubiquitin ligase for MIZ1 (Inoue *et al*, 2013; Yang *et al*, 2010). We show here that MIZ1 globally accumulates at MYC-bound promoters upon inhibition of HUWE1, thereby shifting the equilibrium of MYC complexes from activating toward repressive complexes. shRNA-mediated depletion of MIZ1 attenuated, but did not abolish inhibition of MYC-dependent transactivation by HUWE1 inhibition. It is possible that this is due to the fact that the depletion of MIZ1 is not complete. Alternatively, HUWE1 has additional functions and substrates via which it promotes MYC-dependent transactivation, such as ubiquitination of MYC itself (Adhikary *et al*, 2005). HUWE1 also ubiquitinates and degrades the HDAC2 histone deacetylase, which associates with the MXD/MAX repressive complex (Laherty *et al*, 1997) that antagonizes MYC transactivation, and it is possible that degradation of chromatin-bound HDAC2 is also a critical activity of HUWE1 (Zhang *et al*, 2011).

Notably, depletion of MIZ1 preferentially de-repressed genes that encode ribosomal proteins when HUWE1 was inhibited. Previous work had shown that MIZ1 function is controlled by the ribosomal protein RPL23 that retains nucleophosmin (NPM), a co-activator of MIZ1, in the nucleolus (Wanzel *et al*, 2008). Like HUWE1, NPM antagonizes complex formation of MYC with MIZ1 (Herkert *et al*, 2010). The data reported here therefore argue that HUWE1 and NPM1 are part of a regulatory circuit that couples transcriptional activation of ribosomal protein genes by MYC to the levels of free ribosomal proteins in the nucleolus. Additional circuits exist that couple MYC function to the level of ribosomal proteins (Dai *et al*, 2007).

The notion that HUWE1 antagonizes assembly of MYC/MIZ1 repressive complexes is consistent with genetic data showing that deletion of HUWE1 enhances MYC/MIZ1-dependent repression during skin carcinogenesis (Inoue *et al*, 2013). Importantly, MIZ1-dependent repression of *Cdkn1a* (encoding p21CIP1) expression is a critical function of MYC in *Ras*-driven skin tumorigenesis, since deletion of *MYC* or *MIZ1* inhibits

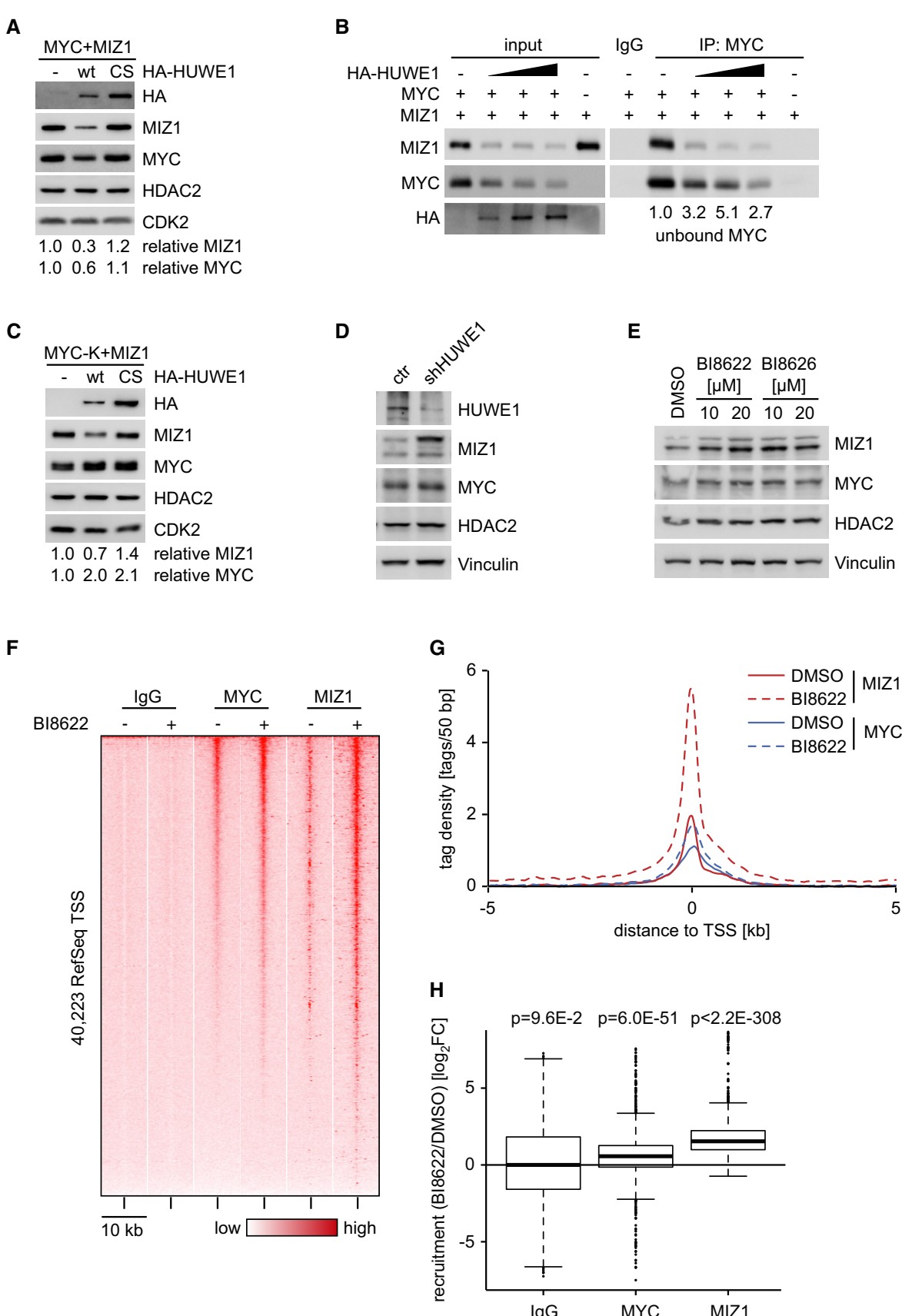

**Figure 6.**

*Ras*-induced tumor formation and upregulates p21CIP1 expression; tumor formation in the absence of MYC or MIZ1 is restored when *Cdkn1a* is co-deleted (Honnemann *et al*, 2012; Oskarsson *et al*, 2006). We propose therefore that HUWE1 degrades MIZ1 in both colon carcinoma cells and keratinocytes, but whether this promotes or inhibits oncogenesis depends on whether transcriptional activation or repression by MYC is critical for oncogenesis in a given tumor.

Surprisingly, neither genetic ablation of HUWE1 (Zhao *et al*, 2008) nor its inhibition (this report) affects expression of MYC target genes in embryonic stem and normal epithelium cells of the intestine, arguing that targeting HUWE1 may open a significant therapeutic window. One factor contributing to this specificity is that embryonic stem cells express both MYC and N-MYC and they are functionally redundant (Smith *et al*, 2010). The association of MIZ1 with N-MYC is weaker than that with MYC (Peukert *et al*, 1997) (E. Wolf and M. Eilers, unpublished observations), and comparison of N-MYC-dependent with MYC-dependent gene expression suggests that N-MYC does not repress transcription via MIZ1 *in vivo* (M. Eilers, unpublished observations). This difference between N-MYC and MYC may explain why N-MYC does not require HUWE1 for transactivation. A second factor contributing to this specificity is the ARF tumor suppressor protein that is expressed in tumor but not in normal cells; ARF inhibits HUWE1, promotes the assembly of MYC/MIZ1 complexes, and inhibits transactivation by MYC (Chen *et al*, 2005; Herkert *et al*, 2010; Qi *et al*, 2004). Additionally, expression of HDAC2 is high in colon carcinoma cells since it is downstream of the WNT pathway (Zhu *et al*, 2004). It is likely, therefore, that the high levels of ARF and HDAC2 enhance the potency of the HUWE1 inhibitors in colon cancer cells and contribute to the specificity of the effects. Collectively, our findings argue that the effects of HUWE1 depletion or inhibition will be dictated by the state of the MYC/N-MYC network and that this will allow specific inhibition of MYC function in colon tumor cells.

Targeting the ubiquitin system is emerging as a possibility to specifically interfere with the assembly and function of oncogenic networks for tumor therapy, and a general inhibitor of the proteasome, bortezomib, is currently in clinical use (Mahindra *et al*, 2012). Proof-of-principle compounds show that it is possible to specifically inhibit individual RING finger and F-box ubiquitin ligase complexes as well as their antagonists, the de-ubiquitinating enzymes (Micel *et al*, 2013). Our data provide strong support for the notion that it is possible to specifically target individual HECT-domain ubiquitin ligases using small molecule inhibitors and use this to control the activity of MYC and possibly other transcription factors for tumor therapy.

# Materials and Methods

### Identification of HUWE1 inhibitors

A DELFIA format assay was used to measure the covalent association of ubiquitin with HUWE1 (auto-ubiquitination). Reactions were carried out on streptavidin-coated 96-well plates (Roche). 10 μl Assay buffer (50 mM Tris–HCl pH 7.5, 50 mM KCl, 0.5 mM $MgCl_2$, 0.05 mM $CaCl_2$, 0.5 mM DTT, 0.1% Tween-20, 1% DMSO) per well is added with 4% DMSO and inhibitor compounds, followed by 20-min incubation at room temperature with 20 μl enzyme mix [3.7 ng His-UBA1, 70 ng His-UbcH5b, 30 ng biotinylated His-HUWE1 (C-terminal ~350 amino acids = HECT-domain), 100 ng His-3xMYC-ubiquitin]. After addition of 10 μl 2 mM ATP, samples were incubated for 3 h at room temperature. 200 μl DELFIA wash buffer (Perkin Elmer) was used each for five washing steps. 50 μl Eu-labeled MYC antibody, diluted 1:5,000 in DELFIA assay buffer (Perkin Elmer), was added followed again by five washing steps. Europium fluorescence was measured with 50 μl DELFIA enhancement solution (Perkin Elmer) in a Wallac Victor fluorescence plate reader (Perkin Elmer).

### Cell culture and transfections

HCT116, HEK293T, HeLa, HT29, ES, Phoenix, and U2OS cells were cultivated in DMEM (Sigma). Ls174T and SW480 cells were grown in RPMI 1640 (Sigma). All media were supplemented with 10% FBS (Biochrom AG) and 1% penicillin/streptomycin (Sigma). Medium for ES cells contained 15% FBS, 1% NEAA (Gibco), 6 μg/ml LIF, and 0.05 mM β-mercaptoethanol (Sigma). For cultivation of ES cells, plates were coated with 0.1% gelatine. PAM212 cells were grown in $Ca^{2+}$-free DMEM (Gibco) supplemented with 10% KGM-2 (Lonza), 10% FBS, 1% L-Glutamine (SAFC Biosciences), 0.8% Gentamycin (Lonza). BI8622 and BI8626 were dissolved in DMSO at a stock concentration of 10 mM and used at 10 or 20 μM final concentration unless indicated otherwise. For UV treatment, cells were exposed to UVB

---

**Figure 7.  Accumulation of MIZ1 contributes to repression of MYC target genes after HUWE1 inhibition.**

A   Concomitant MIZ1 depletion and HUWE1 inhibition in Ls174T cells. Ls174T cells carrying a doxycycline-inducible shRNA targeting MIZ1 were treated with doxycycline (1 μg/ml) for 4 days. At day three, cells were treated with BI8622 (20 μM) for 16 h and MIZ1 protein levels were analyzed in an immunoblot. Vinculin served as loading control.

B   Expression of MYC target genes in MIZ1-depleted Ls174T cells after HUWE1 inhibition. The experiment was carried out as described in (A). Relative expression of indicated MYC target genes was assessed using RQ–PCR. Error bars show SEM of six technical replicates of two biological replicates (*n* = 2).

C   GSEA of RNA sequencing data. *x*-axis shows response of individual gene sets to BI8622, *y*-axis shows response to BI8622 in the presence of shRNA to MIZ1. Red color indicates statistically significant difference in response between control and MIZ1-depeleted cells. Green indicates a gene set mainly or exclusively composed of genes encoding ribosomal proteins.

D   Examples of gene sets encoding ribosomal proteins that are de-repressed upon depletion of MIZ1 in HUWE1-inhibited cells.

E   Heat map demonstrating accumulation of MIZ1 at core promoters of genes encoding ribosomal proteins upon HUWE1 inhibition.

F   Model illustrating our findings. We propose that HUWE1-mediated degradation of MIZ1 is required to prevent accumulation of MYC/MIZ1 complexes at MYC-activated E-boxes in colon carcinoma cells. Degradation of other proteins, such as HDAC2, may contribute to HUWE1-mediated activation of MYC target genes. Finally, ubiquitination of MYC itself by HUWE1 may antagonize binding of MIZ1 (Adhikary *et al*, 2005). Stabilization of MIZ1 upon inhibition of HUWE1 also enhances binding to direct MIZ1 target genes.

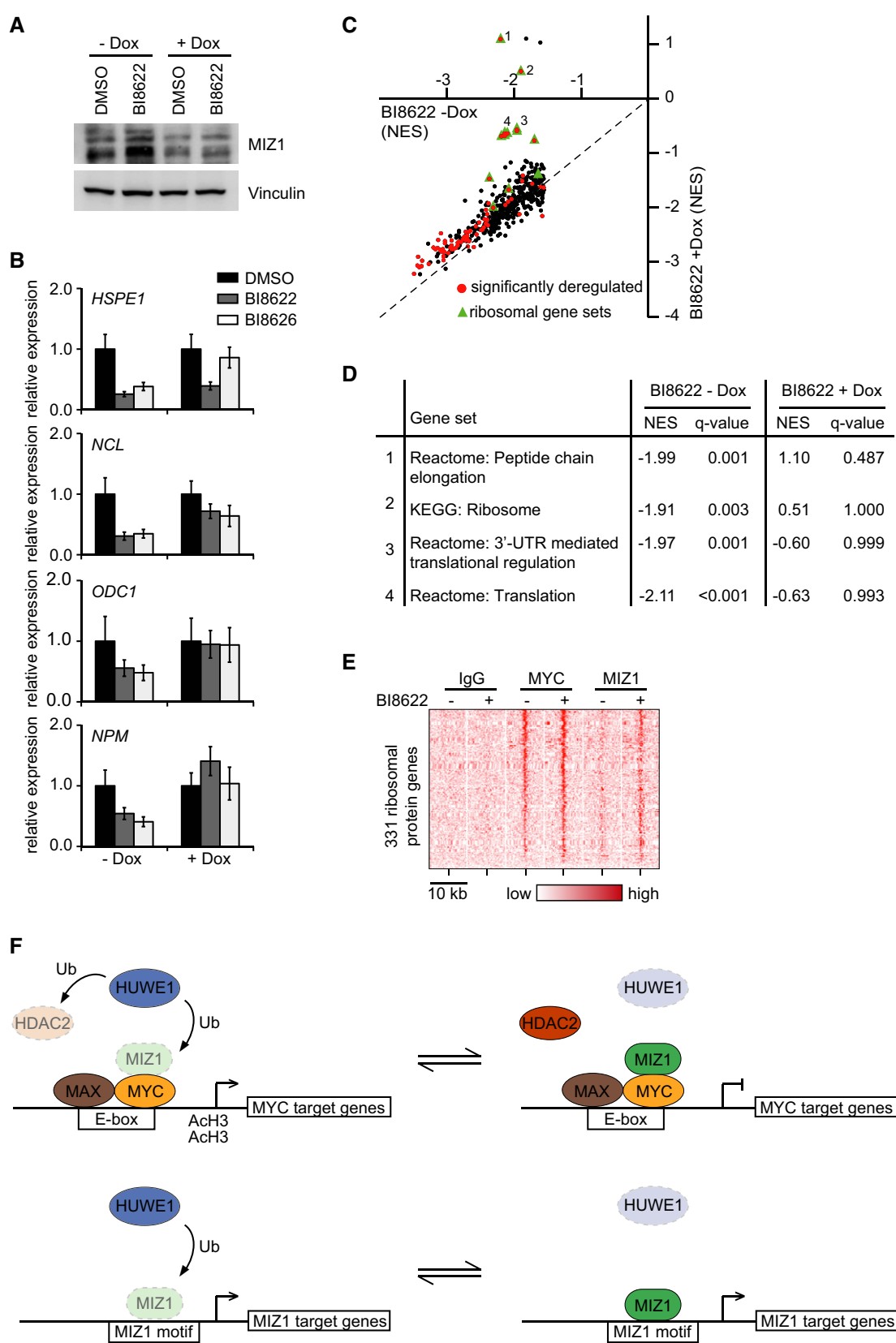

**Figure 7.**

(500 J/m$^2$) for 1 min. Transfections were carried out using PEI (Polyethyleneimine, Sigma). Quantification of crystal violet staining was performed by extracting dye with 10% acetic acid and measuring absorbance at 590 nm. IC$_{50}$ values were calculated using the four parameter logistic equation model.

### Retroviral and lentiviral infections

Retroviruses expressing an shRNA against HUWE1 were generated by transfection of pRetroSuper, pLKO (TRC consortium) or a pInducer (Meerbrey et al, 2011) vector together with psPAX.2 and pMD2.G into HEK293T cells. shRNA sequences are shown in Supplementary Table S1. Details on vector constructs used are available upon request. Cells infected with empty vector or vector expressing non-targeting shRNA were used as controls. Generation of MIZ1-depleted cells included a FACS sorting step.

### Tumor models

Mouse experiments were performed in accordance with Swiss guidelines and approved by the Veterinarian Office of Vaud. Subcutaneous xenografts were established by injection of 500,000 cells into the right flank of NOD/SCID/IL2R common gamma chain deficient (cγ$^{-/-}$) mice under isoflurane anesthesia. Doxycycline (1 mg/ml in drinking water) treatment was started 7 days after engraftment, when tumors became palpable. Tumor progression was monitored by external caliper measurements of the longest (LD) and smallest (SD) diameter of the subcutaneous tumors. For orthotopic xenografts, mice were anesthetized with 90 mg/kg ketamine and 14.5 mg/kg xylazine, and the cecum was exteriorized by laparotomy. 500,000 Ls174T cells were suspended in 15 μl 1/1 matrigel/medium and injected into the cecum. Tumor growth was followed by in vivo imaging of luciferase activity using an IVIS imaging system (Xenogen). For in vivo imaging, mice were injected i.p. with 150 mg/kg luciferine (Biosynth) and imaged for 1–60 s under isoflurane anesthesia. Quantification was performed with LivingImage software (Xenogen). Doxycycline (1 mg/ml in drinking water) treatment was started 14 days after engraftment when the mice showed an abdominal luciferase signal. To assess cell proliferation, mice were injected i.p. with 1 mg BrdU (Sigma) 1 h prior to sacrifice and tumor dissection.

### Intestinal crypt cultures

Murine intestinal crypt cultures were established from wild-type mice of a mixed background (50% C57Bl6J, 50% S129) as previously described (Sato et al, 2009). Briefly, intestines were isolated and flushed with PBS. Villi were removed by scraping and remaining tissue cut and washed with PBS. Crypt extraction was carried out in PBS + 2 mM EDTA for 30 min at 4 °C. Isolated crypts were washed with PBS and plated in growth factor reduced Matrigel (BD Biosciences). Crypts were grown in Advanced DMEM (Invitrogen) supplemented with N2 (Invitrogen), B27 (Invitrogen), EGF (Peprotech), Noggin (Peprotech), and R-spondin (Peprotech). Media were replaced every 2 days and crypts passaged weekly. For drug treatments, crypts were grown until they formed budding "organoids" and then treated with 10 μM BI8622 or control treated with 0.1% DMSO for 24 h. Organoid survival was scored at this point

based on their microscopic appearance. Cell viability was also determined using a CellTiter-Blue Cell Viability Assay (Promega) according to the manufacturer's instructions.

### Immunoblotting and immunoprecipitation

For immunoblotting, cells were lysed in RIPA buffer (50 mM Tris pH 7.5, 150 mM NaCl, 1% NP-40, 0.5% DOC, 0.1% SDS) containing protease inhibitors (Calbiochem). Cleared protein lysates were separated by SDS–PAGE and transferred to a PVDF membrane (Millipore). Immunoprecipitation was performed by lysing cells in IP buffer (20 mM HEPES pH 7.9, 200 mM NaCl, 0.2 mM EDTA, 1% NP-40, 10 mM NaF, 10 mM Na-Pyrophosphate) containing protease inhibitors. Cell lysates were cleared by centrifugation and immunoprecipitated with indicated antibodies. Protein complexes were recovered with Protein A- or Protein G-coupled sepharose beads (Sigma). Immunoprecipitates were washed three times with IP buffer, boiled in SDS sample buffer, and subjected to immunoblotting. Antibodies are listed in Supplementary Table S2.

### Cellular ubiquitination assays

Transfected cells were lysed in 6 M guanidine-HCl, 100 mM phosphate buffer pH 8.0, 10 mM imidazole, 0.4% Triton X-100, 0.125 M NaCl, containing protease inhibitors, 10 μM MG-132 (Sigma), and 5 mM NEM (Sigma). Cell lysates were sonicated and cleared by centrifugation. His-ubiquitin-modified proteins were collected with Ni-NTA-Agarose (Qiagen). After washing, precipitates were boiled in SDS sample buffer supplemented with 200 mM imidazole and subjected to immunoblotting.

### Chromatin immunoprecipitation with High-throughput sequencing (ChIP-seq)

Cells were lysed in ChIP lysis buffer (5 mM PIPES pH 8.0, 85 mM KCl, 0.5% NP-40), and after centrifugation, nuclei were resuspended in ChIP-RIPA buffer (10 mM Tris–HCl pH 7.5, 150 mM NaCl, 1% NP-40, 1% DOC, 0.1% SDS, 1 mM EDTA). DNA was sonicated with a Branson sonifier to obtain DNA fragments ≤ 500 bp. Chromatin was pre-cleared with Protein A- or Protein G-sepharose beads (Sigma, blocked with salmon sperm DNA (Invitrogen) and BSA (Roth)) and immunoprecipitated with indicated antibodies. For ChIP-seq, only BSA was used for blocking. Chromatin/protein complexes were collected by incubation with Protein A- or Protein G-sepharose beads for 6 h. After several washings, chromatin was eluted with 1% SDS/0.1 M NaHCO$_3$ and crosslinking was reverted overnight. DNA was purified with the GeneJet PCR Purification Kit (Thermo Scientific) or by phenol–chloroform extraction and analyzed by RQ–PCR. Percent input was calculated by subtraction of IP or control CT values from input CT values. Primers for RQ–PCR are listed in Supplementary Table S3.

Libraries for ChIP-sequencing were constructed using the NEBNext ChIP-Seq Library Prep Master Mix Set for Illumina (NEB). Briefly, ChIP DNA was end-repaired, A-tailed, Illumina adaptors were ligated, and the DNA separated on a 2% agarose gel. DNA fragments of 200 bps were excised and purified using a QIAquick Gel Extraction Kit (Qiagen). Size-selected DNA was amplified with

18 PCR cycles, quality-controlled using the Experion chip electro-phoresis system (Bio-Rad), and quantified using a picogreen assay. Sequencing of multiplexed DNA libraries was done on an Illumina Genome Analyzer IIx following the manufacturer's instructions. Basecalling was performed using the real-time analysis (RTA) package within the Genome Analyzer Sequencing Control Software (SCS2.10). Demultiplexing and generation of FASTQ files were done with the CASAVA software only considering high quality sequences (PF-cluster).

### RNA sequencing

For RNA sequencing, total RNA was extracted from cells using the RNeasy Mini Kit (Qiagen) with on-column digestion of DNA. Poly-A$^+$ RNA was isolated from total RNA using the NEBNext Poly(A) mRNA Magnetic Isolation Module (NEB). Libraries for RNA sequencing were prepared using the NEBNext Ultra RNA Library Prep Kit for Illumina (NEB) or the NEBNext mRNA Library Prep Master Mix Set for Illumina (NEB) following the instructions of the manufacturer. Depending on the kit used for sample preparation, DNA libraries were size selected using Agencourt AMPure XP Beads (Beckman Coulter) followed by amplification with 12 PCR cycles or by excision of 250 bp fragments from 2% agarose gels and amplification with 13 cycles of PCR. Amplicon sizes and library quantities were determined using the Experion chip electrophoresis system (Bio-Rad). Libraries were sequenced on an Illumina Genome Analyzer IIx following the manufacturer's instructions. Reads were aligned to the human genome (hg19) with BOWTIE v0.12.8 (Langmead, 2010) using default parameters. Analysis of the aligned sequence data was done in R/Bioconductor.

### Microarray analysis

Microarray analyses were performed on a 44K Whole Human Genome Array (G4845A 026652, Agilent), and raw data were generated with the Feature Extraction software v10.1.1.1 (Agilent).

### Statistical analysis and bioinformatic methods

For ChIP-seq experiments, reads were mapped with BOWTIE v0.12.8 (Langmead, 2010) to the human genome (hg19) and peaks were called with IgG sample as control using MACS v1.4.2. The keep-dup parameter was adapted depending on the enrichment at the highest peaks. Peaks were annotated to the nearest RefSeq gene (UCSC GoldenPath RefSeq database) with the "closestBed" feature of the Bedtools suite v2.11.2 (Quinlan & Hall, 2010). Heat maps indicating co-occupancies at transcriptional start sites and corresponding tag density distributions were generated with SeqMiner v1.3.3 in which the strand orientation was taken into account. To calculate recruitment of MYC and MIZ1 after BI8622 treatment, MIZ1-bound promoters (−1 kb to +0.5 kb relative to the TSS) containing consensus E-boxes were selected and the number of tags were counted in a region ±100 bp around the center of the MIZ1 peak. The tool edgeR (Robinson *et al*, 2010) was used to determine differential gene expression and conduct statistical inference.

The *P*-value was calculated using a one-sample two-tailed Student's *t*-test with μ = 0. Gene set enrichment analysis (GSEA)

**The paper explained**

**Problem**

Activation of MYC is a central driver of colorectal carcinogenesis, and genetic experiments argue that inhibition of MYC would have a major therapeutic benefit for this tumor. MYC proteins are transcription factors that interact with their partner proteins and with DNA via large surfaces, making direct targeting with small molecule inhibitors difficult.

**Results**

We had previously shown that transactivation by MYC proteins requires the ubiquitin ligase HUWE1 (HECTH9). We have identified highly specific inhibitors of HUWE1 and found that inhibition of HUWE1 is a feasible approach to inhibit MYC function in a tumor cell-specific manner, since HUWE1 is required to prevent assembly of a repressive complex of MYC with MIZ1 on MYC-activated target genes.

**Impact**

Here, we establish a new principle that allows inhibiting MYC-dependent transactivation for tumor therapy.

was performed using default settings and the C2 gene sets from the MSigDB (www.broadinstitute.org/gsea/msigdb). To compare two GSEA, the normalized enrichment scores (NES) for all repressed gene sets were plotted and gene sets containing > 25% ribosomal protein genes were highlighted. Data are presented as means with standard deviation unless defined differently in the figure legends. *P*-values were calculated using Student's *t*-test.

### Compound stability assays

Human liver microsomes (1 mg/ml, GE Healthcare) were pre-incubated with 10 μM BI8622 or BI8626 in 0.1 M pre-warmed potassium phosphate buffer (pH 7.4) for 5 min at 37 °C in a total volume of 500 μl. To start the reaction, 1 mM NADPH (Sigma) was added and incubated for 0, 5, 7.5, 10, 30, 60 min at 37 °C. Time point 0 min was taken directly after addition of NADPH. At each time point, 50 μl aliquots was taken and the reaction was stopped by addition of 140 μl methanol for subsequent mass spectral analysis. Mice were injected with 10 μl inhibitor compound (10 mM) per gram body weight to achieve a concentration of 100 μM *in vivo*. Mice were sacrificed after 30, 60, 120, and 240 min, and blood samples were taken. Samples were centrifuged (4 °C; 10 min; 800 *g*), and plasma supernatant was used to determine the inhibitor concentration.

### Mass spectrometry

For sample preparation, 50 μl of HUWE1 inhibitor-containing samples (blood plasma or microsomal preparations) was mixed with 10 μl standard solution (50 μM BI8622 or BI8626 in methanol), 140 μl methanol, 150 μl chloroform, and 25 μl 2 M NH$_3$. After centrifugation, the supernatant was mixed with 200 μl chloroform and 50 μl water. The suspension was centrifuged, and the lower phase washed twice with 206 μl of methanol/chloroform/water (100/6/100; v/v/v). After centrifugation, 200 μl of the lower phase was evaporated at 60 °C under a stream of nitrogen gas. The residue was resuspended in 40 μl 1% acetic acid and centrifuged. For mass

spectral analysis, 35 μl supernatant was mixed with 40 μl methanol. Mass spectral data were obtained using an APEX II FT-ICR mass spectrometer.

**Supplementary information** for this article is available online: http://embomolmed.embopress.org

## Acknowledgements

We thank Frauke Debus for initial characterization of the inhibitors, Michael Krause and Lukas Rycak for performing the microarray analysis, Angela Grün, Renate Metz, and Barbara Bauer for excellent technical assistance, and members of the Eilers laboratory for comments on the manuscript. We thank Michel Aguet (EPFL Lausanne) for hosting JB in his laboratory during this project. This work was supported by the Wilhelm Sander-Stiftung for Cancer Research and a grant from the Boehringer Ingelheim Fonds (to LAJ).

## Author contributions

SP, JB, KM, LAJ, SW, JM, MG, MT, GB, CPA, WS, AW, and CO performed the experiments; CPA, SP, JB, and SW analyzed data; NP, OS, NK, and ME designed experiments, and ME wrote the paper.

## Conflict of interest

The authors declare that they have no conflict of interest.

## For more information

Accession code for the microarray experiments: E-MTAB-1890

See also: http://www.pch2.biozentrum.uni-wuerzburg.de/startseite/

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
