## [Review Process File · EMBO Molecular Medicine]

Tumor cell-specific inhibition of MYC function using small molecule inhibitors of the HUWE1 ubiquitin ligase

Stefanie Peter, Jennyfer Bultinck, Kevin Myant, Laura A. Jaenicke, Susanne Walz, Judith Müller, Michael Gmachl, Matthias Treu, Guido Boehmelt, Carsten P. Ade, Werner Schmitz, Armin Wiegering, Christoph Otto, Nikita Popov, Owen Sansom, Norbert Kraut, and Martin Eilers

Corresponding author: Martin Eilers, Theodor Boveri Institute

Review timeline:

Submission date:	01 February 2014
Editorial Decision:	05 February 2014
Appeal:	05 February 2014
Additional Editorial Correspondence:	06 February 2014
Editorial Decision:	19 March 2014
Revision received:	30 July 2014
Accepted:	26 August 2014

Transaction Report:

Editor: Roberto Buccione

1st Editorial Decision

05 February 2014

Thank you for the submission of your manuscript "Tumor cell-specific inhibition of MYC function using small molecule inhibitors of the HUWE1 ubiquitin ligase".

I have now had the opportunity to carefully read your paper and the related literature and I have also discussed it with my colleagues and an external expert. I am afraid that we concluded that the manuscript is not well suited for publication in EMBO Molecular Medicine and have therefore decided not to proceed with peer review.

You follow up on your previous findings that Myc proteins need to be ubiquitinated the ubiquitin ligase Huwe1 to be active, by designing, synthesizing and testing new potential small molecule inhibitors of Huwe1.

Although we acknowledge the potential interest of your findings, we feel that they are a bit preliminary at this stage. The lack of a thorough characterisation of compound selectivity (e.g. vs. the many other ligases) and the relatively high IC50 in cells, is such that your work appears more suited to a specialistic venue at this stage. Also, direct experimentation in suitable animal models would also be required down the line to reach the translational significance we would like to see in an EMBO Molecular Medicine article.

I am sorry that I could not bring better news.

Thank you very much for your rapid reply. May be not surprisingly, we feel that your reply does not do justice to the vast amount of work, which went into this manuscript. In our view, this work is not "a bit preliminary". Could I ask you to briefly consider the following arguments? You raise three main points:

(1) Lack of specificity: When performing the compound screen, our colleagues at Boehringer were well aware that all proteins that contain active cysteine can be targeted by non-specific electrophiles. This is why they included several controls: most importantly, they screened a total of 10 hect-domain ubiquitin ligases, all of which contain an active cysteine: of these, only Huwe1 was inhibited (Figure 3C). They also tested the E1/E2-enzymes, both of which also contain an active cysteine, separately, and they were also not inhibited (described on page 7). So there is no way to escape the conclusion that the compounds target specifically Huwe1. The lack of effect on other Hect domain ligases is also implicit in other data: for example, we show that the compounds have no effect on Myc-target genes in Huwe1-depleted cells (Figure 5A) and the comparison of compound data with shRNA data in an unbiased gene expression analysis (Figure 5F) rules out a non-specific effect. In this respect, the compounds are far better characterized than multiple inhibitors that are currently being used. The manuscript contains multiple additional controls for specificity: for example, the compounds have no effect on p53 levels in HeLa cells, in which p53 is turned over by the Hect-domain E3 ligase E6-AP. Finally, the ES cell experiments rule out non-specific cell toxicity. We believe, therefore, that the respective statement in your e-mail does not accurately reflect the multiple controls we included to ensure specificity.

(2) IC50 values: Here, the data situation is similar to the recently published Usp7 inhibitors, which also belong to a class of closely related cysteine proteases. These were published in *Cancer Cell* (22, 345-358) (also in *Molecular Cell Biology* 33, 3309-3320). The key evidence for the specificity of these inhibitors is the same as we used: counter-screening against several other closely related Usp7 demonstrates specificity for Usp7. Importantly, the concentrations used in the Usp7 papers are higher than those we use here and the biochemical effects are at best similar to those we report. On a broader note, multiple papers (including papers published in *Cell*) now use "Myc-inhibitors" at much higher concentrations. Even the I-BET inhibitors used to target expression of the MYC and MYCN promoters that are in preclinical development show single digit EC50 values for growth inhibition of MYCN-amplified neuroblastoma cells (see: Figure 1E of Puissant, *Cancer Discovery* March 2013 3; 308). Finally, the concentrations we use are virtually identical to the concentrations used for the new Skp2 F-box ubiquitin ligase inhibitor published in *Cell* last year (Chan, *Cell* 154, 556, 2013). We believe, therefore, that the IC50 values are within the range of many experimental and preclinical inhibitors.

(3) In vivo validation: The central open issue was whether Huwe1 can be validated in an in vivo setting as a therapeutic target, since this was open from the genetics published so far. Of course we realize that the manuscript lacks in vivo data for the compounds and we are trying this experiment. However, the pharmacokinetic properties of the compounds appear not optimal and they appear to be turned over by cytochrome P450, so we need to test different dosing and formulation schemes. The results are currently open. But the data shown in Figure 1, as they stand, fully validate Huwe1 as a target for colon carcinoma in vitro, in subcutaneous and in orthotopic assays. As such, they provide a large progress relative to the state of the field, in particular since they are compatible with the published genetic data. They provide a clear path forward for further optimization of these compounds for therapy.

After hundreds of attempts, there is still no significant way to inhibit Myc oncogenic function and most attempts rely on "Myc-inhibitors" that bind the Myc leucine zipper with zero demonstration of specificity whatsoever. Proof-of-principle papers that use a dominant-negative transgene document that if you could inhibit Myc, this would have tremendous therapeutic effects. These purely conceptual papers have been published by *Nature*, *Genes and Development* and similar journals (see: Soucek and Evan for references). The data we show here establish two key novel principles

First, they show that it is possible to specifically target Hect-domain E3 ligases and second, they show a unique and novel way of targeting Myc function. We would therefore like to politely ask you to reconsider your editorial decision. I will be happy to discuss these issues with you on the phone, should that be helpful. If so, please indicate a time, when you would be available.

Additional Editorial Correspondence

06 February 2014

Thank you again for your mail of yesterday.

Dr. Buccione has rediscussed your paper with his colleagues and the external expert and has consequently decided to send your manuscript out for peer review.

2nd Editorial Decision

19 March 2014

Thank you for the submission of your manuscript to EMBO Molecular Medicine. We are very sorry that it has taken so long to get back to you on your manuscript.

In this case we experienced unusual difficulties in securing three willing and appropriate reviewers. Further to this, we are still missing one evaluation. As a further delay cannot be justified I have decided to proceed based on these available consistent evaluations.

Both Reviewers find merits in your manuscript although they raise significant issues that require your action. I will not dwell into much detail as their comments are detailed. I would like, however, to highlight a few main points.

Reviewer 2, while recognising the broad interest of your work, contends that the data supporting a role for Miz1 as potential effector of MYC activity in a HUWE1-less background do not fully justify the conclusions and suggests further experimentation to solidify the point. S/he also notes that the consequences of HUWE1 over-expression on c-MYC target gene expression should be tested in colorectal cancer cells. Finally, Reviewer 2 mentions that the main claim as indicated by the title, requires further experimental support.

Reviewer 3 is also globally positive, but notes that the therapeutic potential of the compounds targeting HUWE1 requires further rigorous experimentation aimed at verifying a number of crucial aspects including pharmacokinetics, in vivo efficacy and convincing proof that the compounds do target HUWE1, thus in part matching Reviewer 2's comments. As does Reviewer 2, this Reviewer goes into significant detail to explain his/her points and suggests possible experimental avenues.

When interrogated during the cross-commenting process, one Reviewer stressed the point that your work is very interesting and does compellingly show that HUWE1 E3 ubiquitin ligase is a target for malignancies with MYC involvement but that identification of small molecules that ostensibly selectively target HUWE1 requires significant additional support. The Reviewer also suggested that you should be given the opportunity to address the above concerns. I completely agree and therefore, while publication of the paper cannot be considered at this stage, we would be prepared to consider a substantially revised submission, with the understanding that the Reviewers' concerns must be addressed with additional experimental data where appropriate and that acceptance of the manuscript will entail a second round of review. I understand that if you do not have the data available at least in part, this would represent a significant amount of work. I would thus encourage you to develop your study as far as realistically possible for your next, revised version to strengthen your findings and increase their impact; in this respect, if you could provide in vivo proof of

concept, I would not require you to provide pharmacokinetics data too.

Since as mentioned above, the required revision in this case appears to require a significant amount of time, additional work and experimentation and might be technically challenging, I would therefore understand if you chose to rather seek publication elsewhere at this stage. Should you do so, we would welcome a message to this effect.

Please note that it is EMBO Molecular Medicine policy to allow a single round of revision only and that, therefore, acceptance or rejection of the manuscript will depend on the completeness of your responses included in the next, final version of the manuscript.

As you know, EMBO Molecular Medicine has a "scooping protection" policy, whereby similar findings that are published by others during review or revision are not a criterion for rejection. However, I do ask you to get in touch with us after three months if you have not completed your revision, to update us on the status. Please also contact us as soon as possible if similar work is published elsewhere.

***** Reviewer's comments *****

Referee #2 (Remarks):

In their manuscript, Peter et al. show that downregulation of the E3-ligase HUWE1 inhibits tumor growth and metastasis by colorectal cancer cell. Furthermore, they characterize novel small molecule inhibitors directed against HUWE1 activity. These inhibitors mimic RNAi-mediated ablation of HUWE1. At the molecular level, the authors observe reduced expression of c-MYC-activated target genes upon ablation of HUWE1 in LS174T cells whereas c-MYC protein levels remain unaffected. Apparently, ablation of HUWE1 leads to stabilization of Miz1 protein and increased recruitment of Miz1 to the c-MYC-activated target gene HSPE1. Authors conclude that accumulation of c-MYC/MAX/Miz1 complexes at c-MYC activated promoters might be responsible for the observed molecular and phenotypic changes induced by HUWE1 down-regulation or chemical inhibition in colorectal cancer cells. Finally, since HUWE1 ablation does neither inhibit c-MYC-target gene trans-activation nor cell growth in mouse embryonic stem cells, the authors claim that the effects observed must be tumor cell-specific.

The effect of HUWE1 on colorectal cancer cell growth and the specificity of the novel small molecule inhibitors towards HUWE1 function are convincing. However, data supporting a role for Miz1 as potential effector of MYC activity in a background of HUWE1 ablation do not justify authors' conclusions. The mechanism proposed is of broad interest yet some aspects should be addressed more carefully. I detail my criticisms below;

Major points:

1.) The group of Martin Eilers published already several years ago (Adhikary et al., 2005) that HUWE1-mediated ubiquitination of c-MYC plays an important role for its trans-activating function whereas c-MYC protein stability was not affected. According to the present manuscript, mutating the HUWE1 target sites within c-MYC impairs c-MYC-mediated gene activation and also its function *in vivo*. Authors propose that stabilization of Miz1 and co-localization to c-MYC target gene promoter(s) is responsible for HUWE1 inhibition-mediated changes of the c-MYC transcriptome and on cell growth. Yet, data presented are not compelling. To further support the proposed model, authors should show that ablation of Miz1 rescues the effects on HUWE1 downregulation, i.e. Authors should perform a concomitant knock-down of HUWE1 and Miz1 in colorectal cancer cell lines.

2.) Localization of Miz1 to c-MYC target gene promoters after HUWE1 inhibition is demonstrated for one single case (HSPE1). From this, the authors conclude that Miz1-recruitment to c-MYC target gene promoters explains, at least partially, the compromised transactivation of c-MYC target genes upon HUWE1 inhibition culminating in reduced tumor growth and metastasis. As the suggested mechanism can be of broad interest to the Myc and the cancer community, authors should address to which extent increased Miz1 abundance at c-MYC promoters correlates with altered c-MYC gene expression after HUWE1 inhibition using a genome wide approach.

3.) Over-expression of HUWE1 is shown to cause a drop in Miz1 levels thereby increasing the amount of MYC that is not bound to Miz1. No experiments were done describing the consequences of HUWE1 over-expression on c-MYC target gene expression in colorectal cancer cells. This experiment is particularly relevant as Adhikary et al. reported previously that HUWE1 is over-expressed in primary tumors and has a positive impact on tumor growth.

4.) The title of the manuscript highlights tumor-cell specific effects on c-MYC function induced by HUWE1 inhibition. However, this conclusion is only drawn from the observation that HUWE1 ablation does not negatively impact on mouse ES cell growth. Authors should provide additional evidences to support this notion. For instance, in vitro models of normal mouse and human colonic mucosa are readily available and authors could test the effects of HUWE1 chemical or genetic inhibition on the growth of normal colonic stem cells.

Referee #3 (Remarks):

Comments for Authors: EMM-2014-03927-V2-Q

Eilers and colleagues provide an interesting series of studies that suggest that targeting the HUWE1 E3 ubiquitin ligase (also known as MULE and ARF-BP1) is an attractive strategy to disable malignancies having MYC involvement (here colorectal cancers).

Using genetic and new small molecule inhibitors the authors show that HUWE1 contributes to the growth and tumorigenic potential of colorectal cancer cells, and that mechanistically this is due to: (i) its role as a required coactivator of Myc induced genes; and (ii) its ability to direct the destruction of MIZ1, which represses MYC targets. In effect this is a double-whammy on transcriptional programs directed by Myc. Based on a convincing series of studies the authors propose the reasonable model that HUWE1 is required for Myc transactivation functions due to its ability to prevent assembly of repressive MYC/MAX/MIZ1 complexes at target genes that are induced by MYC.

In some respects these findings are rather predictable given previous work from this group and that of Tak Mak showing a similar response in other tumor cell types (Inoue et al., *Genes & Dev.*, 2013; Adhikary et al., *Cell*, 2005). The novelty here is: (1) convincing studies that support the model that HUWE1 inhibits the formation of MYC-MIZ1 complexes; and (2) the development of small molecule therapeutics that target HUWE1, which is viewed as a potentially important advance. However, a more rigorous evaluation of such small molecules as therapeutics is warranted, and the mechanistic studies presented also need to be further developed, for this body of work to be appropriate for consideration in EMBO Molecular Medicine.

Major Concerns:

(1) Rigor needs to be applied in evaluating the therapeutic potential of the small molecule HUWE1 inhibitors described in this study. Indeed, no convincing data are presented to indicate that these small molecules would have a suitable therapeutic window in the clinic, nor that they have anti-

cancer potency *in vivo*. The fact that ES cells are not affected by these HUWE1 targeting agents is not surprising - one can generate and grow c-Myc-deficient ES cells. What are needed here is: (i) a rigorous assessment of the effects of these small molecules on normal epithelial cells *ex vivo*; (2) *in vivo* efficacy studies; (3) DMPK analyses of such agents to determine if their drug-like properties are suitable for starting points for lead compounds; and (4) rigorous evaluation of the potential toxicity of such compounds - hematological parameters, liver enzymes/blood chemistry, tissue analyses, etc.

(2) Additional experiments are needed to demonstrate that the small molecule inhibitors that are in hand are indeed on target. First, HUWE1 overexpression and knockdown studies should shift the EC50 of these compounds (which have modest M potency). Second, the authors should perform experiments that prove that these agents indeed only selectively bind to and inhibit HUWE1 in cells. For example, a chemical tag could be added to the top two compounds and proteins that bind to the compound confirmed by mass spec analyses following pull-down.

(3) The authors claim that continuous degradation of MIZ1 is necessary for the response observed following HUWE1 inhibition is not supported by any data. This conclusion begs for the experiment where the authors test whether a mutant form of MIZ1 that cannot be ubiquitinated by HUWE1 is sufficient to confer resistance to their small molecule HUWE1 inhibitors, and to silencing of HUWE1. In addition the authors need to formally show that silencing and overexpression of HUWE1 indeed affects the half-life of MIZ1, by providing pulse-chase immunoprecipitation analyses.

(4) In addition to MIZ1 antagonism, Myc transcription functions are antagonized by the related bHLH-Zip transcription factors Mnt and Mad1-Mad4. What are the effects of HUWE1 on this regulatory circuit?

Minor Points:

1. Paragraphs need to be indented.
2. Figure 2A. Contrary to the statement made it looks as though silencing of HUWE1 does have effects on the steady state levels of c-Myc protein, which look lower in these cells.
3. What are the effects of HUWE1 silencing on the localization of c-Myc? This issue may have been dealt with in previous studies but if so its not stated.
4. What are the effects of HUWE1 silencing on dimerization of Myc with Max? Do the K63 ubiquitin chains directed by HUWE1 affect dimerization? DNA binding? Again if these facts are known they should at least be stated.
5. It is hard to envision that a Myc protein devoid of lysine residues is a functional protein.

Response to Reviewer' comments

We would like to thank both referees for their insightful and helpful comments to our manuscript. Please find below a description of the changes that we introduced in reply to these comments.

Referee #2

1.) The group of Martin Eilers published already several years ago (Adhikary et al., 2005) that HUWE1-mediated ubiquitination of c-MYC plays an important role for its trans-activating function whereas c-MYC protein stability was not affected. According to the present manuscript, mutating the HUWE1 target sites within c-MYC impairs c-MYC-mediated gene activation and also its function in vivo. Authors propose that stabilization of MIZ1 and co-localization to c-MYC target gene promoter(s) is responsible for HUWE1 inhibition-mediated changes of the c-MYC transcriptome and on cell growth. Yet, data presented are not compelling. To further support the proposed model, authors should show that ablation of MIZ1 rescues the effects on HUWE1 downregulation, i.e. Authors should perform a concomitant knock-down of HUWE1 and MIZ1 in colorectal cancer cell lines.

Several changes in the revised manuscript address this comment:

First, we mention in the revised Introduction results described in a manuscript that has now been published in Nature showing that in tumor cells there is a balance of activating (MYC/MAX) and repressive (MYC/MAX/MIZ1) complexes on virtually all target genes of MYC and that the balance between both complexes found on each gene determines the response to MYC (Walz et al., 2014).

Second, we have performed a ChIP-Seq experiment showing that MIZ1 globally accumulates on MYC-target genes after inhibition of HUWE1 (New Figures 6 F,G,H) (also see response to point 2 below).

Third, we have performed the requested experiment and inhibited HUWE1 in the presence of an shRNA targeting MIZ1. Depletion of MIZ1 retards cell proliferation by itself, therefore we cannot expect a rescue of proliferation. The level of depletion that we achieved is documented in Figure 7A. At this level of depletion, RQ-PCR assays of individual genes show that depletion of MIZ1 alleviates, but does not completely abolish repression upon HUWE1 inhibition (Figure 7B). We have therefore performed a genome-wide analysis by RNA-sequencing. This shows that depletion of MIZ1 does not affect all MYC-target genes uniformly. Strikingly, it abolishes repression upon HUWE1 inhibition on target genes encoding ribosomal proteins (Figure 7C,D). We mention in the revised discussion that the new data fit extremely well to previous observations from Rosalie Sears' and our group that show that both MYC and MIZ1 function are controlled by regulatory circuits that monitor levels of free ribosomal proteins.

2.) Localization of MIZ1 to c-MYC target gene promoters after HUWE1 inhibition is demonstrated for one single case (HSPE1). From this, the authors conclude that MIZ1-recruitment to c-MYC target gene promoters explains, at least partially, the compromised transactivation of c-MYC target genes upon HUWE1 inhibition culminating in reduced tumor growth and metastasis. As the suggested mechanism can be of broad interest to the MYC and the cancer community, authors should address to which extent increased MIZ1 abundance at c-MYC promoters correlates with altered c-MYC gene expression after HUWE1 inhibition using a genome wide approach.

In the revised manuscript, we have replaced the analysis of *HSPE1* with a genome-wide analysis of MYC and MIZ1 occupancy after HUWE1 inhibition (see new Figure 6 F,G,H). The data show a strong global increase in MIZ1 levels at target genes of MYC, causing a global shift from MYC/MAX to MYC/MAX/MIZ1 complexes. There is also a very small increase in

MYC, consistent with multiple data demonstrating that association with MIZ1 stabilizes MYC. This is small but relevant since it shows that MYC is not exported from the nucleus and, since DNA binding to E-boxes depends on MAX, MYC remains bound to MAX after HUWE1 inhibition (also see below).

3.) Over-expression of HUWE1 is shown to cause a drop in MIZ1 levels thereby increasing the amount of MYC that is not bound to MIZ1. No experiments were done describing the consequences of HUWE1 over-expression on c-MYC target gene expression in colorectal cancer cells. This experiment is particularly relevant as Adhikary et al. reported previously that HUWE1 is over-expressed in primary tumors and has a positive impact on tumor growth.

Stable overexpression of the full-length HUWE1 protein - at 500kD - is as far as we know impossible; we have tried all technologies and vectors available. Transient overexpression does not result in significant changes in MYC-dependent gene expression.

4.) The title of the manuscript highlights tumor-cell specific effects on c-MYC function induced by HUWE1 inhibition. However, this conclusion is only drawn from the observation that HUWE1 ablation does not negatively impact on mouse ES cell growth. Authors should provide additional evidences to support this notion. For instance, in vitro models of normal mouse and human colonic mucosa are readily available and authors could test the effects of HUWE1 chemical or genetic inhibition on the growth of normal colonic stem cells.

We have performed RQ-PCR assays to show that inhibition of HUWE1 does not affect MYC target genes in ES cells (Figure 4F).

We have also performed extensive experiments in crypt cultures of normal colonic epithelial cells. We do not observe any effect of HUWE1 inhibition on expression of MYC target genes in multiple independent assays and we include this new result in Figure 4G. Upon long-term incubation of the extremely sensitive primary crypt cultures, a fraction of the individual crypts died. This number was tolerable for BI8622 (around 30%), but high for BI8626 (60-80%). This effect, which may be on-target (since HUWE1 has a role in base excision repair), precluded long-term analyses of crypt growth. When corrected for the number of live crypts at the end of the experiment, MTT assays showed no reduction by BI8622, arguing that BI8622 does not inhibit proliferation of these cells (see Figure for Reviewer at the end of these comments, panel A). Consistent with this interpretation, crypts remain Ki67-positive after incubation with BI8622, although it is virtually impossible to put a precise number to this due to variations in staining intensity and in localization of Ki67-positive cells (See Figure for Reviewers, panel B). In order to take these results into account, we have re-worded the Summary to reflect precisely our observations: "Using high throughput screening we identify small molecule inhibitors of HUWE1, which inhibit MYC-dependent transactivation in colorectal cancer cells, but not in stem and normal epithelial cells."

Referee #3

Major Concerns:

(1) Rigor needs to be applied in evaluating the therapeutic potential of the small molecule HUWE1 inhibitors described in this study. Indeed, no convincing data are presented to indicate that these small molecules would have a suitable therapeutic window in the clinic, nor that they have anti-cancer potency in vivo. The fact that ES cells are not affected by these HUWE1 targeting agents is not surprising - one can generate and grow c-MYC-deficient ES cells. What are needed here is: (i) a rigorous assessment of the effects of these small molecules on normal epithelial cells ex vivo; (2) in vivo efficacy studies; (3) DMPK

analyses of such agents to determine if their drug-like properties are suitable for starting points for lead compounds; and (4) rigorous evaluation of the potential toxicity of such compounds - hematological parameters, liver enzymes/blood chemistry, tissue analyses, etc.

We have done multiple experiments to address this comment.

First, we have used mass spectrometry to determine the stability of HUWE1 inhibitors *in vitro* in the presence of microsomes. These experiments showed a rapid turnover of both compounds with a half-life of 32 and 9.5 minutes, respectively. This is now shown in Figure E7C and mentioned on page 8.

Second, we have measured whether the compounds accumulate after intraperitoneal injection *in vivo*. We injected them at a level that is expected to generate a final concentration of 100 μ M in the mouse. The maximum concentration that we ever found in plasma is around 2 μ M for BI8622 and 8 μ M for BI8626 and these levels further decrease rapidly within the time frame we would need to detect visible changes in gene expression (Figure E7D, page 9). Consistently, we did also not see signs of toxicity in a routine blood analysis (not shown). We concluded that both inhibitors are too unstable to further characterize them *in vivo*. Not mentioned is that the mice also showed signs of severe pain, so we really could not continue these experiments.

(see also reply to point 4 of reviewer #2): Third, we have also performed extensive experiments in crypt cultures of normal colonic epithelial cells. We do not observe any effect of HUWE1 inhibition on expression of MYC target genes in multiple independent assays and we include this new result in Figure 4G, confirming that they specifically inhibit MYC function in colon tumor cells (We also include a new Figure 4F, which shows MYC-dependent gene expression in ES cells). Upon long-term incubation of the extremely sensitive primary crypt cultures, a fraction of the individual crypts died. This number was tolerable for BI8622 (around 30%), but high for BI8626 (60-80%). This effect, which may be on-target, precluded long-term analyses of crypt growth. When corrected for the number of live crypts at the end of the experiment, MTT assays showed no reduction by BI8622, arguing that BI8622 does not inhibit proliferation of these cells (see Figure for Reviewers at the end of these comments, panel A). Consistent with this interpretation, crypts remain Ki67-positive after incubation with BI8622, although it is virtually impossible to put a precise number to this due to variations in staining intensity and in localization of Ki67-positive cells (See Figure for Reviewers, panel B). In order to take these results into account, we have re-worded the Summary to reflect precisely our observations: "Using high throughput screening we identify small molecule inhibitors of HUWE1, which inhibit MYC-dependent transactivation in colorectal cancer cells, but not in stem and normal epithelial cells."

(2) Additional experiments are needed to demonstrate that the small molecule inhibitors that are in hand are indeed on target. First, HUWE1 overexpression and knockdown studies should shift the EC50 of these compounds (which have modest μ M potency). Second, the authors should perform experiments that prove that these agents indeed only selectively bind to and inhibit HUWE1 in cells. For example, a chemical tag could be added to the top two compounds and proteins that bind to the compound confirmed by mass spec analyses following pull-down.

Stable overexpression of the full-length HUWE1 protein - at 500kD - is as far as we know impossible; we have tried all technologies and vectors available. In contrast, shRNA-mediated depletion of HUWE1 is possible. A complete depletion of HUWE1 would arrest cells by themselves, so we used cells with a partial depletion of HUWE1. This causes a moderate, but highly reproducible (n=4) decrease in EC₅₀ values and we include the respective experiment as Figure E6.

Unfortunately, we really do not have the capacity and/or resources to chemically synthesize derivatives of the inhibitors. To address whether the effects on MYC function are on-target, we have now compared the effects of inhibitors to depletion of HUWE1 on gene expression

in an unbiased manner. We add an additional panel to Figure 5 (5G) that shows a global analysis documenting that inhibition of HUWE1 has no significant effect on MYC-dependent gene expression in HUWE1-depleted cells. This allows us to conclude that the effects on MYC are clearly an on-target effect of the drug on HUWE1. Since Myc-dependent changes are by far the major effects on gene expression, we can conclude that virtually all effects on gene expression are on target. From this analysis, we cannot completely exclude that the inhibitors have off-target effects that do not translate into changes in gene expression.

(3) The authors claim that continuous degradation of MIZ1 is necessary for the response observed following HUWE1 inhibition is not supported by any data. This conclusion begs for the experiment where the authors test whether a mutant form of MIZ1 that cannot be ubiquitylated by HUWE1 is sufficient to confer resistance to their small molecule HUWE1 inhibitors, and to silencing of HUWE1. In addition the authors need to formally show that silencing and overexpression of HUWE1 indeed affects the half-life of MIZ1, by providing pulse-chase immunoprecipitation analyses.

We suggest in our model that degradation of MIZ1 is a critical function of HUWE1, so a stable allele of MIZ1 would be a dominant inhibitor of MYC-dependent transactivation. We would argue that the critical test of our model is an experiment in which HUWE1 inhibition is combined with depletion of MIZ1. The results of this experiment are described in the reply to Referee #2, point 1 and shown in the revised Figure 7.

We have also performed multiple MIZ1 stability assays and they show experimental variations that upon averaging do not give good enough results for publication. To address this issue, we have therefore included a new experiment that shows that ectopic expression of HUWE1 decreases MIZ1 levels and that this is blocked by both inhibition of HUWE1 and by the addition of the proteasome inhibitor, MG132 (Figure E11A).

(4) In addition to MIZ1 antagonism, MYC transcription functions are antagonized by the related bHLH-Zip transcription factors Mnt and Mad1-Mad4. What are the effects of HUWE1 on this regulatory circuit?

We have been able to detect Mnt, Mxd1, Mxd3 and Mxd4 (now the official name of MAD proteins) in colon carcinoma cells and there is no effect of either HUWE1 depletion or inhibition on the levels of these proteins. We now mention this result on page 11 and show it in Figure E9A.

Minor Points

1. Paragraphs need to be indented.

This has been done in the revised version.

2. Figure 2A. Contrary to the statement made it looks as though silencing of HUWE1 does have effects on the steady state levels of c-MYC protein, which look lower in these cells.

We have not observed consistent effects on levels of MYC protein and have therefore replaced the panel by a more representative one. The ChIP-sequencing experiments argue there is a small increase in chromatin-bound MYC, consistent with the fact that association with MIZ1 stabilizes MYC.

3. What are the effects of HUWE1 silencing on the localization of c-MYC? This issue may have been dealt with in previous studies but if so its not stated.

4. What are the effects of HUWE1 silencing on dimerization of MYC with MAX? Do the K63

ubiquitin chains directed by HUWE1 affect dimerization? DNA binding? Again if these facts are known they should at least be stated.

We have added two figures showing the respective experiments as Extended Data (E9B and E10). There is no significant effect on localization of MYC or association with MAX.

5. It is hard to envision that a MYC protein devoid of lysine residues is a functional protein.

We have extensively characterized this protein since it turns out to have exciting gene regulatory properties. These data will be the main topic of a different manuscript. Relevant for this story is that the DNA binding properties (localization on chromatin) and ability to activate transcription in reporter assays of K-less MYC are indistinguishable from normal MYC, so we believe it is a valid comparison.

Figure for Reviewers

A.

The plot shows MTT assays after 24 hr incubation with 10 μ M BI8622 or solvent control. At the end of the experiment, the number of life crypts was counted by visual inspection. It was on average 90 % of all crypts for control cultures and 68 % for BI8622- treated cultures (n=3). MTT values are plotted relative to this number.

B.

Pictures show Ki67-staining of intestinal crypts after 24 hr incubation with 10 μ M BI8622 or DMSO as solvent control.

Thank you for the submission of your revised manuscript to EMBO Molecular Medicine. We have now received the enclosed reports from the reviewers who were asked to re-assess it. As you will see the reviewers are now globally supportive and I am pleased to inform you that we will be able to accept your manuscript pending the following final amendments:

1) Every published paper now includes a 'Synopsis' to further enhance discoverability. Synopses are displayed on the journal webpage and are freely accessible to all readers. They include a short standfirst - to be written by the editor - as well as 2-5 one sentence bullet points that summarise the paper (to be written by the author). Please provide the short list of bullet points that summarise the key NEW findings. The bullet points should be designed to be complementary to the abstract - i.e. not repeat the same text. We encourage inclusion of key acronyms and quantitative information. Please use the passive voice. Please attach these in a separate file or send them by email, we will incorporate them accordingly.

2) As you know, as part of the EMBO Publications transparent editorial process initiative EMBO Molecular Medicine will publish online a Review Process File to accompany accepted manuscripts. This file will be published in conjunction with your paper and will include the anonymous referee reports, your point-by-point responses and all pertinent correspondence relating to the manuscript, including figures to referees. If you do NOT want this file to be published or part of it (such as the figures for Reviewers) , please let me know.

3) We are now encouraging the publication of source data, particularly for electrophoretic gels and blots, with the aim of making primary data more accessible and transparent to the reader. Would you be willing to provide a PDF file per figure that contains the original, uncropped and unprocessed scans of all or at least the key gels used in the manuscript? The PDF files should be labeled with the appropriate figure/panel number, and should have molecular weight markers; further annotation may be useful but is not essential. The PDF files will be published online with the article as supplementary "Source Data" files. If you have any questions regarding this just contact me.

Please provide the above information within two weeks (the earlier the better!).

***** Reviewer's comments *****

Referee #2 (Remarks):

Authors have addressed my main criticisms satisfactorily, which have strengthened the manuscript. This is a relevant work and of broad interest for the cancer community.

Referee #3 (Remarks):

Eilers and colleagues provide a remarkably improved study where the authors now convincingly show that stabilization and binding of Miz1 to Myc target genes directs HUWE1-mediated inhibition of Myc transcription.

The authors have addressed all of the concerns in a thorough and rigorous fashion and the new data presented (in Figures 4F-G, 5G, 6F-H, 7C-D, E7C-D, E9A-B, E10, E11A) fully support the authors'

conclusions. Some of the new data are very compelling particularly those showing that Miz1 knockdown abolished the effects on ribosomal gene targets following depletion of HUWE-1 (7C-D). Other major additions are (i) HUWE-1 controls turnover of Miz1; (ii) data showing that the effects of HUWE-1 are selective for the MYC-Miz1 circuit; (iii) a genome-wide assessment of the Miz1 and Myc occupancy following depletion of HUWE-1; (iv) data showing that HUWE-1 depletion does not affect MYC localization and MYC:MAX interaction; and (v) data showing that the small molecule HUWE-1 inhibitor does not affect normal intestinal crypts.

All of the new data presented are convincing. Overall this is viewed as an outstanding study that makes a major contribution to the field.